# Changes in ferrous iron and glutathione promote ferroptosis and frailty in aging *Caenorhabditis elegans*

**Nicole L Jenkins[1], Simon A James[2], Agus Salim[3,4,5], Fransisca Sumardy[1], Terence P Speed[4,6], Marcus Conrad[7], Des R Richardson[8], Ashley I Bush[1]\*, Gawain McColl[1]\***

[1]Melbourne Dementia Research Centre, Florey Institute of Neuroscience and Mental Health and University of Melbourne, Parkville, Australia; [2]Australian Synchrotron, ANSTO, Clayton, Australia; [3]Department of Mathematics and Statistics, La Trobe University, Bundoora, Australia; [4]Bioinformatics Division, Walter and Eliza Hall Institute of Medical Research, Parkville, Australia; [5]Melbourne School of Population and Global Health, and School of Mathematics and Statistics, University of Melbourne, Melbourne, Australia; [6]Department of Mathematics and Statistics, University of Melbourne, Melbourne, Australia; [7]Helmholtz Zentrum München, Institute of Metabolism and Cell Death, Neuherberg, Germany; [8]Department of Pathology and Bosch Institute, University of Sydney, Sydney, Australia

**\*For correspondence:**
ashley.bush@florey.edu.au (AIB);
gmccoll@florey.edu.au (GMC)

**Abstract** All eukaryotes require iron. Replication, detoxification, and a cancer-protective form of regulated cell death termed *ferroptosis*, all depend on iron metabolism. Ferrous iron accumulates over adult lifetime in *Caenorhabditis elegans*. Here, we show that glutathione depletion is coupled to ferrous iron elevation in these animals, and that both occur in late life to prime cells for ferroptosis. We demonstrate that blocking ferroptosis, either by inhibition of lipid peroxidation or by limiting iron retention, mitigates age-related cell death and markedly increases lifespan and healthspan. Temporal scaling of lifespan is not evident when ferroptosis is inhibited, consistent with this cell death process acting at specific life phases to induce organismal frailty, rather than contributing to a constant aging rate. Because excess age-related iron elevation in somatic tissue, particularly in brain, is thought to contribute to degenerative disease, post-developmental interventions to limit ferroptosis may promote healthy aging.

## Introduction

Life has evolved to exploit the redox chemistry of iron for essential activities. Ferrous iron drives ferroptosis, a regulated cell death program genetically and biochemically distinct from apoptosis, necrosis and autophagic cell death (*Dixon et al., 2012*). Ferroptosis kills malignant cells but may also be inappropriately activated in ischemic injury and neurodegeneration (*Stockwell et al., 2017*; *Tuo et al., 2017*; *Viswanathan et al., 2017*; *Yang and Stockwell, 2016*). This cell death mechanism is executed by (phospho)lipid hydroperoxides induced by either iron-dependent lipoxygenases, or by an iron-catalyzed spontaneous peroxyl radical-mediated chain reaction (autoxidation). Under homeostatic conditions the ferroptotic signal is terminated by glutathione peroxidase-4 (GPx4), a phospholipid hydroperoxidase that needs glutathione as a cofactor. While the signaling that regulates ferroptosis has been studied in depth (*Conrad et al., 2016*; *Doll et al., 2017*; *Jiang et al., 2015*), the role of iron load in this death signal is poorly resolved (*Doll and Conrad, 2017*).

Redox cycling between $Fe^{2+}$ and $Fe^{3+}$ can contribute to cellular stress. This is mitigated by a range of storage and chaperone pathways to ensure that the labile iron pool is kept to a minimum (*Hare et al., 2013*). In *Caenorhabditis elegans* the emergence of labile ferrous iron with age correlates with genetic effects that accelerate aging (*James et al., 2016*; *James et al., 2015*) and could be a lifespan hazard (*Xu et al., 2010*). Excess iron supply has been shown to shorten lifespan in *C. elegans* (*Gourley et al., 2003*; *Hu et al., 2008*), yet variable results have been reported with iron chelation. The iron chelator deferiprone was reported not to impact *C. elegans* lifespan (*Valentini et al., 2012*), but this study was limited by indirect measures of iron load, use of only a single dose of deferiprone, and small sample size. In contrast, use of calcium-ethylenediaminetetraacetic acid (CaEDTA), a non-specific chelator that does not redox-silence iron, caused a minor (undisclosed) increase in lifespan (*Klang et al., 2014*). Whether selective targeting of ferrous iron burden can impact on aging and lifespan is unknown.

The developmental dependence on iron for reproduction and cellular biochemistry may represent an ancient and conserved liability in late life. The load of tissue iron increases needlessly in aging nematodes, mammals and humans (*James et al., 2015*; *Massie et al., 1983*; *Massie et al., 1985*; *Ward et al., 2014*). This must tax regulatory systems that prevent abnormal redox cycling of iron, such as the $Fe^{2+}$-glutathione complexes thought to be the dominant form of iron in the cellular labile iron pool (*Berndt and Lillig, 2017*; *Hider and Kong, 2011*). We hypothesized that age-dependent elevation of labile iron, coupled with a reduction of glutathione levels conspire to lower the threshold for ferroptotic signaling, increasing the vulnerability of aged animals and implying that disruption to the iron-glutathione axis is fundamental to natural aging and death. To test this, we investigated the vulnerability to ferroptosis of aging nematodes upon the natural loss of glutathione during lifespan. We examined the effects of inhibiting ferroptosis in *C. elegans* using two distinct treatments: a potent quenching agent for lipid peroxidation (autoxidation) (*Zilka et al., 2017*) as well as a small lipophilic iron chelator (*Kalinowski and Richardson, 2005*) that prevents the initiation and amplification of lipid peroxide signals. Our analysis of these interventions provides mechanistic insight into the influence of the $Fe^{2+}$-glutathione couple on the occurrence of natural ferroptosis across lifespan, at the cellular and organismal level. Because excess age-related iron elevation in somatic tissue, particularly in brain (*Massie et al., 1983*; *Massie et al., 1985*; *Ward et al., 2014*), is thought to contribute to degenerative disease (*Hare et al., 2013*; *Hare et al., 2015*), our data indicate that postdevelopmental interventions to limit ferroptosis may promote healthy aging.

## Results

### Glutathione depletion vulnerability

Glutathione is suggested to be the dominant coordinating ligand for cytosolic ferrous iron (*Hider and Kong, 2011*) and is also the substrate used by glutathione peroxidase-4 (GPX4) to clear the lipid peroxides that induce ferroptotic cell death (*Dixon and Stockwell, 2014*; *Friedmann Angeli et al., 2014*; *Yang et al., 2014*). Deletion of four *C. elegans* homologs of GPX4 decreases lifespan (*Sakamoto et al., 2014*), but whether ferroptosis mediates this is unknown. We tested whether acute depletion of glutathione can initiate ferroptosis in adult *C. elegans* using diethyl maleate (DEM), which conjugates glutathione (*Urban et al., 2017*; *Valmas and Ebert, 2006*). DEM has been reported to produce a non-linear response to glutathione depletion, with a minor glutathione loss induced by DEM at 10–100 µM increasing lifespan via hormesis, but a major glutathione loss induced by DEM $\geq 1$ mM shortening lifespan (*Urban et al., 2017*). We found that DEM $\geq 1$ mM induced death in 4 day old adult worms (at the end of their reproductive phase) in a dose- and time-dependent manner, with $\approx 50\%$ lethality occurring after 24 hr exposure to 10 mM DEM (*Figure 1A*) associated with $\approx 50\%$ depletion of glutathione (*Figure 1B*). We also found that total glutathione levels steadily decrease with normal aging, approaching $\approx 50\%$ on Day 10 of the levels on Day 1 (*Figure 1C*, *Supplementary file 1*). This may contribute to *C. elegans* becoming disproportionately more vulnerable to DEM lethality as they enter the midlife stage (*Figure 1D*).

We tested whether lethality associated with glutathione depletion was caused by ferroptosis. We examined the treatment of *C. elegans* with the selective ferroptosis inhibitor, liproxstatin (Lip-1, 200 µM) (*Dixon et al., 2012*). We also targeted the accumulation of late life iron (*James et al., 2016*; *James et al., 2015*), that catalyses (phospho)lipid hydroperoxide propagation, using salicylaldehyde

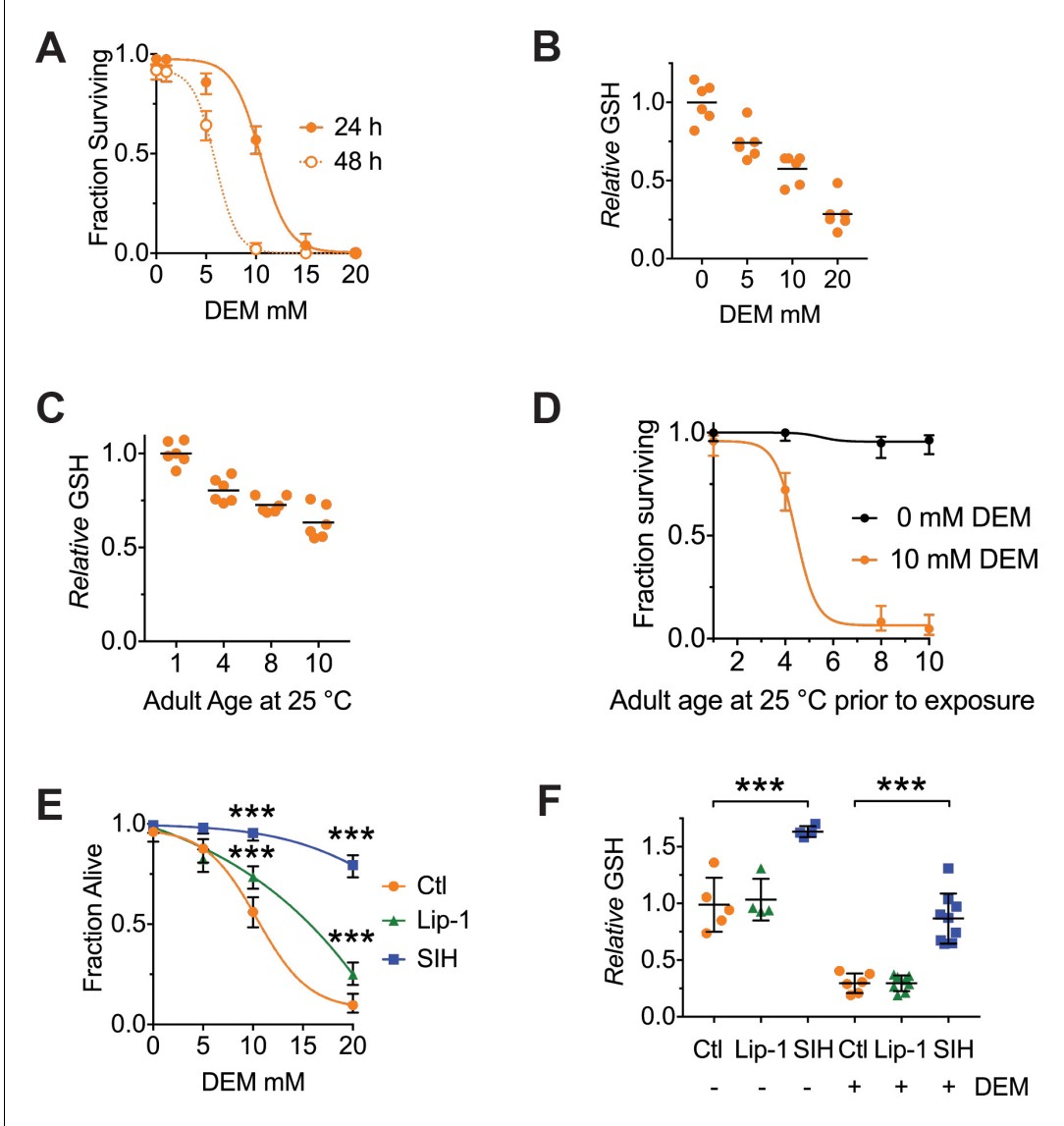

**Figure 1.** Both Lip-1 and SIH protect against toxicity from glutathione depletion. Treatment with DEM represents an acute stress that reduces glutathione levels and causes death, with older animals being more susceptible.      Note, the age of adults is determined by the number of days following the last larval molt and therefore reflects the number of days of adulthood. (A) Survival curves of adults following either 24- or 48 hr exposure to increasing doses of DEM. Treatment begun on Day 4 of adulthood. Shown are proportions ± 95% confidence intervals (*Newcombe, 1998*), with a sigmoidal curve fitted. (B) Total glutathione (GSH) decrease following 6 hr of DEM exposure. Day 4 adults used, with results normalized to the GSH levels in worms not exposed to DEM. Plotted are six independent replicates, with each estimated derived from 50 adults per measure). Linear regression $R^2$ = 0.98, p=0.01 (C) Total GSH levels decrease with increased adult age in *C. elegans*. Each point is derived from six independent replicates of 50 adults, with black lines marking the mean value. Results are normalized to the GSH levels in Day one worms (ANOVA: F (3, 20)=32.96, p<0.0001; see *Supplementary file 1* for pairwise comparisons). (D) Aged *C. elegans* adults become progressively more sensitive to GSH depletion by DEM. Shown are proportions ± 95% confidence intervals, with a sigmoidal curve fitted. (E) Both Lip and SIH treatment protect against lethality from DEM-derived glutathione depletion. Day 4 adults, with values representing pooled data from four independent experiments ± 95% confidence intervals, each with a fitted sigmoidal curve. Ctl denotes vehicle control (0.5% v/v DMSO). Pairwise comparisons at 10 and 20 mM DEM were performed using Fisher's exact test; *** denotes p<0.001. (F) Total glutathione levels are preserved following SIH pretreatment from L4, but not by Lip-1. Day 4 adults were exposed to DEM (10 mM) for 6 hr, and total glutathione (GSH) assayed. Ctl denotes vehicle control (0.5% v/v DMSO). Each point is derived from 4 to 9 independent replicates of 50 adults, with black lines marking the mean value ± SD. (ANOVA: F (5, 30)=50.97, p<0.0001; see *Supplementary file 1* for pairwise comparisons). *** denotes p<0.001.

The online version of this article includes the following source data for figure 1:

**Source data 1.** Data for DEM and GSH comparisons.

isonicotinoyl hydrazone (SIH, 250 μM), a lipophilic acylhydrazone that scavenges intracellular iron and mobilizes it for extracellular clearance (*Kalinowski and Richardson, 2005*). Importantly, unlike chelators such as CaEDTA, iron bound by SIH does not redox cycle (*Chen et al., 2018*). For both interventions, *C. elegans* were treated from early adulthood (late L4) onwards to eliminate any potential developmental effects.

DEM toxicity in 4-day-old worms was rescued by both Lip-1 and SIH (*Figure 1E*, *Supplementary file 1*), with more marked protection by SIH. This is consistent with ferroptosis contributing to the death mechanism. Therefore, the fall in glutathione with aging (*Figure 1C*, *Supplementary file 1*) would be expected to interact synergistically with the concomitant rise in labile iron (*James et al., 2016*; *James et al., 2015*) to increase the risk of ferroptosis. We found that this age-dependent rise in iron itself may contribute to the fall in glutathione, since pretreatment of the worms with SIH from L4 prevented the age-dependent decrease in glutathione when assayed on Day 4 of adult life (*Figure 1F*). Furthermore, SIH mitigated the glutathione depletion induced by DEM in Day four animals (*Figure 1F*), demonstrating that cytosolic iron synergizes the depletion of glutathione initiated by DEM. While Lip-1 alleviated the lethality of DEM (*Figure 1E*), it did not prevent the fall in glutathione that was induced by aging (as assayed on Day 4) or by DEM (*Figure 1F*). Thus, Lip-1 inhibition of ferroptosis in *C. elegans* occurs downstream of glutathione depletion, consistent with its effect in rescuing ferroptosis in cultured cells (*Shah et al., 2018*).

## Individual cell ferroptosis heralds organismal demise

A feature of ferroptosis is the propagation of cell death in a paracrine manner mediated by uncertain signals that might include the toxic lipid peroxidation end-products 4-hydroxynonenal (4-HNE) and malondialdehyde (MDA) (*Feng and Stockwell, 2018*; *Linkermann et al., 2014*). Compared to strong oxidants like the hydroxyl radical, 4-HNE and MDA are relatively stable and able react with macromolecules, such as proteins distal to the site of origin. To determine whether individual cell death precedes organismal death in our model of aging, we used propidium iodide to visualize moribund cells in vivo after DEM treatment and during aging. Propidium iodide (PI) is a fluorescent intercalating agent that binds to DNA, but cannot cross the membrane of live cells, making it possible to identify the nuclei of recently dead or dying cells, as shown in *Figure 2A* (and *Figure 2—figure supplement 1*). Examination of aged cohorts, or young animals treated with DEM, indicated that cell death (particularly death of intestinal cells) preceded organismal death in both 4-day-old (*Figure 2B*) and 6- and 8-day-old (*Figure 2C*) adults, and was significantly attenuated by both Lip-1 or SIH. Death of individual cells prior to organismal death is consistent with progressive degeneration contributing to the frailty phenotype.

The PI-positive dying cells did not accumulate with aging (*Figure 2—figure supplement 1*), perhaps because the dead cells are cleared during the remaining lifespan of the animal. It is known that as *C. elegans* ages, intestinal nuclei are lost (*McGee et al., 2011*) and the propidium iodide cannot stain nuclei if they are absent. Additionally, we would not expect a linear increase proportional to age in the prevalence of animals with stained cells during longitudinal studies of our cohorts, because dead animals are removed from the population and the rate of death changed over time for the cohorts (Figure 5C; see below). Thus, the prevalence of PI-positive cells per animal would be a complex product of the rate of PI emergence, the rate of PI disappearance, the rate of nuclear disappearance and the rate of organismal death. However, we were able to survey the prevalence of animals with any dead cells on particular days in the adult life span. This determined that cell death begins to be detected after 4 days of age, and that our interventions with SIH and Lip-1 completely suppressed this cell death at 6 and 8 days of age (*Figure 2C*).

To estimate changes in lipid peroxidation, we assayed MDA via the thiobarbituric acid reactive substance assay. As expected, acute glutathione depletion by DEM exposure caused a marked increase in the relative amounts of MDA (*Figure 2D*). We also observed an aged-related increase in MDA, consistent with an age-related increase in ferroptotic signalling in *C. elegans*, that was ameliorated by both Lip-1 and SIH treatment (*Figure 2E*). Consistent with the MDA results, we also found a concomitant qualitative increase in 4-HNE protein adducts with age that was suppressed by both Lip-1 and SIH treatments (*Figure 2F*).

We considered whether the higher levels of glutathione in animals treated with SIH (*Figure 1E and F*) could reflect a hormetic response to sublethal oxidative stress, which has been described for SIH at low concentrations (10 μM) combining with the cellular labile iron pool within hepatocellular

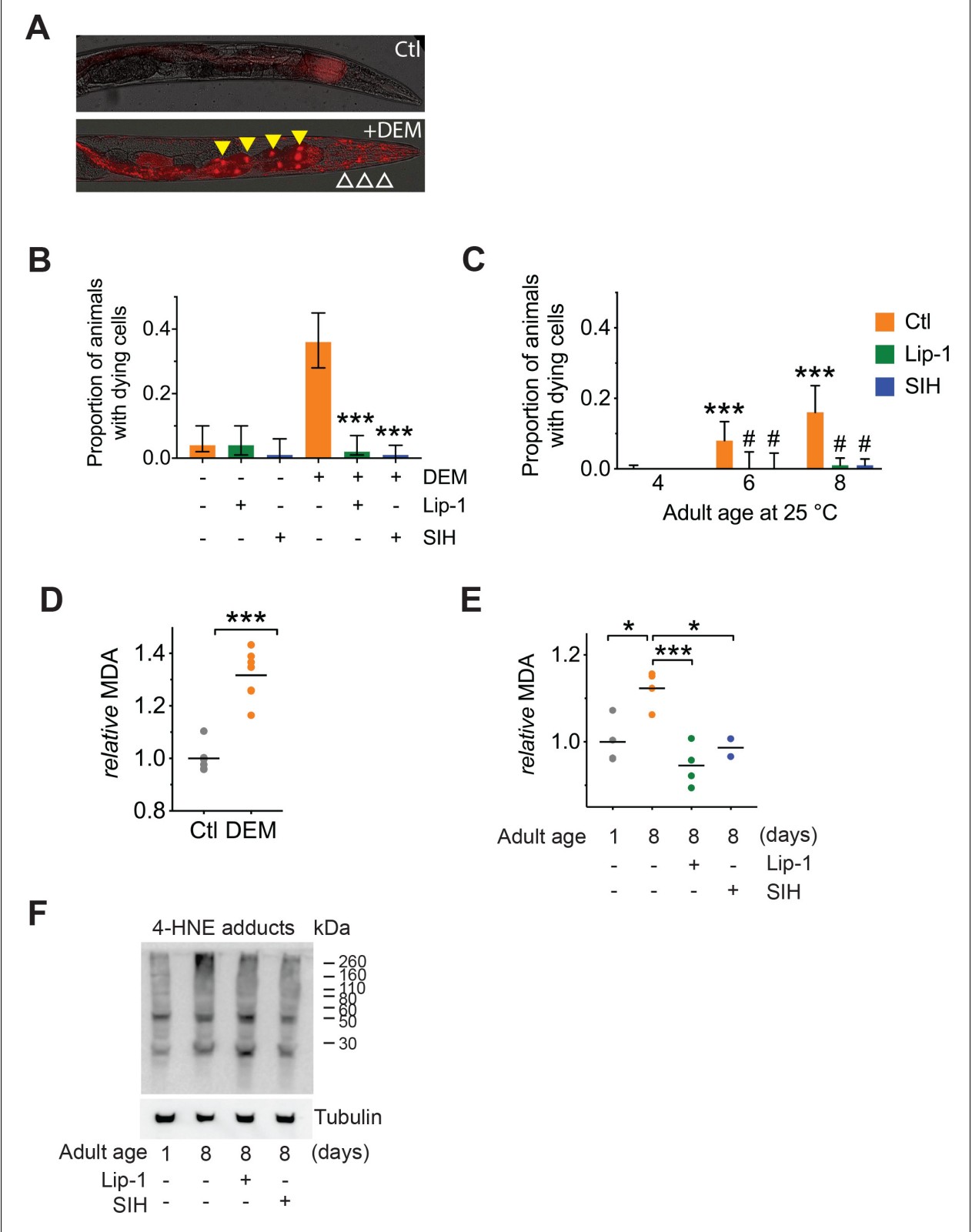

**Figure 2.** |Both Lip-1 and SIH inhibit cell death and protect against lipid peroxidation. Measures of lipid peroxidation and cell death show an increase with age and reduction by both Lip-1 and SIH treatment.     In all panels, vehicle control (0.5% v/v DMSO, Ctl)-treated worms are shown in orange, Lip-1-treated (200 μM Lip-1) are green and SIH-treated (250 μM SIH) are blue. (**A**) Representative propidium iodide fluorescence (red) overlay of bright field micrograph depicting dead intestinal cells (marked by nuclear signal, yellow triangles) within a live Day 4 adult treated with DEM. Smaller fluorescent
*Figure 2 continued on next page*

Figure 2 continued

puncta were also observed, consistent with neuronal cell nuclei (white unfilled triangles). Untreated Day 4 adult control animals (Ctl) showed no cell death. (B) Proportion of live animals at Day 4 (±95% confidence intervals) of adulthood showing dead cell fluorescence (propidium iodide)±exposure to 10 mM DEM for 24 hr. Cohorts of animals included: co-treatment with a vehicle control (-DEM, Ctl, n = 102; +DEM Ctl n = 117), Lip-1 (-DEM, Lip-1, n = 84; +DEM Ctl n = 106) or SIH (-DEM, Ctl, n = 89; +DEM Ctl n = 129). Lip-1 and SIH both markedly reduced the proportion of animals with dead cells after DEM treatment (z-test: Ctl vs Lip-1 Z = 6.37 ***p<0.001; Ctl vs SIH Z = 7.24, ***p<0.001). (C) Proportion of live animals at 4, 6 and 8 days of adulthood (±95% confidence intervals) showing propidium iodide nuclear fluorescence. Day 6 Ctl adults (n = 160) had a significantly higher proportion of animals with dead cells than Day 4 Ctl (n = 263; z-test: Z = 4.70 ***p<0.001). Similarly, Day 8 Ctl adults (n = 119) had significantly higher prevalence of animals with propidium iodide nuclear fluorescence than Day 4 Ctl (z-test: Z = 6.65 ***p<0.001). No significant difference was observed between Day 6 and Day 8 control populations. Within the Day 6 adult cohorts treatment with Lip-1 (n = 77) or SIH (n = 81) markedly reduced the prevalence of animals with propidium iodide nuclear fluorescence compared to vehicle control (z-test: Ctl vs Lip-1 Z = 2.57 # p<0.001; Ctl vs SIH Z = 2.64, # p<0.001). Within the Day 8 adult cohorts treatment with Lip-1 (n = 234) or SIH (n = 308) markedly reduced the prevalence of animals with propidium iodide nuclear fluorescence compared to vehicle control (Ctl, n = 119; z-test: Ctl vs Lip-1 Z = 5.67 # p<0.001; Ctl vs SIH Z = 6.28, # p<0.001). (D) Levels of the lipid peroxidation end product malondialdehyde (MDA) increases in *C. elegans* following acute glutathione depletion by 20 mM DEM exposure for 6 hr. MDA levels are shown as values normalized against the mean of untreated Day 4 Adults (Ctl) for independent samples. (Ctl vs +DEM, unpaired 2-tailed t-test ***p<0.001) (E) Malondialdehyde (MDA) increases in aged *C. elegans* (Day 1 vs Day 8 adults, ANOVA *p<0.05). Treatment with either Lip-1 (Day 8 vs Day 8 +Lip-1 adults, ANOVA ***p<0.001) or SIH (Day 8 vs Day 8 +SIH adults, ANOVA *p<0.05) reduces levels of MDA. Data represent independent samples with values normalized against the mean of untreated Day 1 adults. (F) Representative immunoblot against 4-HNE protein adducts comparing Day 1 and Day 8 control adults and aged adults treated with Lip-1 and SIH with corresponding tubulin blot below (representative of triplicate experiments). The relative intensity of the bands show an age-related increase that is ameliorated by Lip-1 and SIH.

The online version of this article includes the following source data and figure supplement(s) for figure 2:

**Source data 1.** Data for cell death and lipid peroxidation.
**Figure supplement 1.** Propidium iodide staining: Cell death observed following treatment with DEM.
**Figure supplement 2.** Propidium iodide staining: Cell death observed in aging nematodes.
**Figure supplement 3.** DAF-16 nuclear localization: The DAF-16 reporter strain TJ356 (zIs356 [Pdaf-16::daf-16a/b::gfp + rol-6(su1006)] was used to visualize nuclear localization of the DAF-16 transcription factor as an indicator of insulin-like signalling.

carcinoma cells in culture (*Caro et al., 2015*). The decrease we observed in our oxidation markers, MDA and 4-HNE, by SIH treatment at 250 µM in *C. elegans* (*Figure 2E and F*) suggests that this higher dose of SIH was sufficient to debulk reservoirs of total iron. To further discount possible off-target stress responses elicited by our interventions, we interrogated DAF-16 localization. Nuclear localization of the DAF-16 transcription factor is an indicator of insulin-like signalling, which occurs under stress conditions (*Henderson and Johnson, 2001*). Neither 250 µM SIH nor 200 µM Lip-1 induced DAF-16 nuclear translocation. As a positive control, treatment with 10 mM DEM did induce nuclear localization of DAF-16 (*Figure 2—figure supplement 2*), consistent with this challenge inducing acute stress. Taken together, these findings argue against hormesis mediating the benefits of SIH or Lip-1 under these conditions in *C. elegans*.

## Changes in iron quantity, speciation and cytoplasmic fraction

Lowering cellular iron suppresses ferroptosis, but the peroxyl radical trapping ferroptosis inhibitors, such as Lip-1, are not expected to change iron levels. We examined the impact of SIH and Lip-1 interventions on iron levels over lifespan using synchrotron-based X-ray fluorescence microscopy (*Ganio et al., 2016*; *James et al., 2015*) to measure both iron concentration (presented as areal density, pg $\mu m^{-2}$) and total (pg per worm) iron (*Figure 3A*; *Supplementary file 7*). Both total iron and areal density increased with age in control animals (*Figure 3B and C*; *Supplementary file 2*), as expected (*James et al., 2015*). SIH dramatically reduced the areal density of iron (and reduced variance) with aging (*Figure 3C*; *Supplementary file 2*), but Lip-1 did not alter iron density. Notably, by Day 8, animals treated with SIH contained total iron load on par with the untreated control group (*Figure 3C*; *Supplementary file 2*), as the lower areal density was offset by an increase in body size of SIH-treated worms compared to age matched controls. These results highlight how bulk measures of total iron or measurements by inference (*Valentini et al., 2012*) can be confounded by changes in the animal morphology when exploring aging interventions.

We had previously determined age-related changes to the *C. elegans* iron-proteome, characterized on size exclusion chromatography by three major peaks: a high molecular weight peak (HMW, >1 MDa), ferritin, and a low MW peak (LMW, 600 Da) that may contain labile iron. With aging, iron redistributes in *C. elegans* out of the ferritin peak (where it is sequestered in redox-silent storage

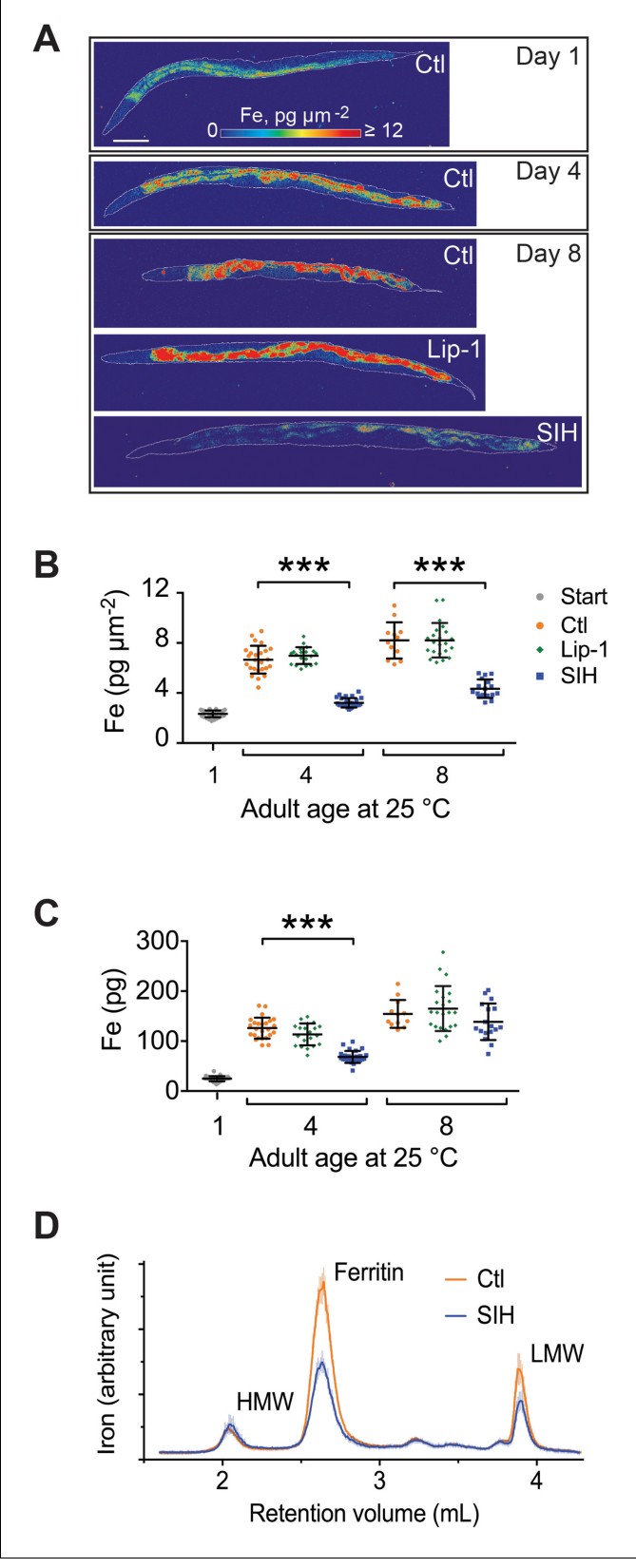

**Figure 3.** Effects of Lip-1 and SIH on iron levels and distribution in *C. elegans*. In all panels, vehicle control (0.5% v/v DMSO, Ctl) treated worms are shown in orange, Lip-1 treated (200 µM Lip-1) are green and SIH treated (250 µM SIH) are blue. (**A**) Representative X-ray fluorescence microscopy maps of tissue iron (Fe) reported as areal

*Figure 3 continued on next page*

*Figure 3 continued*

density (pg µm$^{-2}$) for a first day adult (Day 1) and animals treated for four days (Day 4) and eight days (Day 8) with vehicle control (Ctl), Lip-1 or SIH at 25°C. Scale bar = 50 µm. (**B**) Plot of mean areal density for iron (pg µm$^{-2}$) for all treatment cohorts aged at 25°C. The starting population (Day 1, vehicle control (0.5% v/v DMSO); Start)) is shown in grey. The control cohort (orange) shows an age related increase in total iron (as previously observed [*James et al., 2015*]). The Lip-1 group (green) has similar iron levels across each age, whereas the SIH cohort (blue) has markedly less total iron (ANOVA: $F_{(6,148)}=171.3$, $p<0.0001$; see *Supplementary file 2* for sample summary and pair-wise comparisons). Each data point represents a value from a single *C. elegans* adult, with mean ± SD, ***$p<0.001$. (**C**) Plot of total body iron (pg) for treated *C. elegans* cohorts aged at 25°C. Each data point represents a value from a single *C. elegans* adult, with mean ± SD. All treatments have increased total iron across age with SIH-treated (blue) worms retaining significantly less iron than control (red) and Lip-1 (green) treated worms at Day 4 (ANOVA: $F_{(6,148)}=97.3$, ***$p<0.0001$; see *Supplementary file 2* for sample summary and pair-wise comparisons) (**D**) Native, size-exclusion chromatography of iron-macromolecular complexes from 10 day old adults treated with vehicle control (Ctl, shown in orange) or SIH-treated cohorts (shown in blue). The means ± SD, from three independent biological replicates, are plotted. The three major peaks include unaltered high molecular weight complexes (HMW,>1 MDa,~2.2 mL retention volume), ferritin bound iron (~2.7 mL retention volume; previously identified as FTN-2 [*James et al., 2015*]; area under the peak decreased by ~53% relative to Ctl) and low molecular weight iron complexes (LMW,<30 kDa,~3.9 mL retention volume, decreased ~47% relative to Ctl).

The online version of this article includes the following source data for figure 3:

**Source data 1.** Data for XFM and mass spectrometry comparisons.

reserves) and accumulates in the HMW and LMW peaks (*James et al., 2015*). The chromatographic profile of aged *C. elegans* (10 days post adulthood) treated with SIH (*Figure 3D*) revealed decreased iron associated with the LMW peak (normalized peak area approximately 40%). Ferritin-bound iron was also similarly decreased by SIH (normalized peak area approximately 50%), but iron bound within HMW species was unaffected. The age-related changes in LMW iron are consistent with increased labile iron, which is withdrawn as the substrate for ferroptosis by SIH treatment.

## Fe$^{2+}$ increase with aging is normalized by liproxstatin and SIH

X-ray absorption near edge structure (XANES) spectroscopy, using fluorescence detection for visualization, directly assesses the in vivo coordination environments of metal ions in biological specimens (φXANES) (*James et al., 2016*). The centroid of the XANES pre-edge feature reflects the relative abundance of ferrous [Fe$^{2+}$] and ferric [Fe$^{3+}$] species (*Westre et al., 1997*). Since Fe$^{2+}$ in the labile iron pool is the specific substrate for ferroptosis, and rises with aging in *C. elegans* (*James et al., 2015*), we investigated the impact of our interventions using φXANES (*James et al., 2016*). This synchrotron-based spectroscopy allowed us to evaluate steady state iron speciation (Fe$^{2+}$/ Fe$^{3+}$) in a specific region (anterior intestinal, *Figure 4A*; *Figure 4—figure supplement 1*) of intact, cryogenically-stabilized control, Lip-1 and SIH-treated worms. We found that the age-related increase in the Fe$^{2+}$ fraction was normalized to that of a young animal by both Lip-1 and SIH treatments (*Figure 4B and C*; *Figure 4—figure supplement 2A and B*; *Supplementary file 3*).

Higher levels of pro-ferroptotic Fe$^{2+}$ might be compounded by a loss of glutathione. So, we also assessed changes in fractional Fe$^{2+}$ induced by lethal glutathione depletion by DEM. φXANES of 4 day old wild type worms treated with DEM identified a marked increase in the Fe$^{2+}$ fraction (*Figure 4D*; *Figure 4—figure supplement 2C & D*; *Supplementary file 3*), revealing the upper limit for tolerable Fe$^{2+}$ fraction being about 0.3 of the total iron (*Figure 4D*). These results help to contextualize the observed increase in Fe$^{2+}$ during normal aging also being about 0.3 of the total iron (*Figure 4C*), which was normalized to ≈0.2 by Lip-1 or SIH intervention.

## Lifespan effects of ferroptosis inhibition or blocking iron accumulation

Since Fe$^{2+}$ accumulates with aging and contributes to *C. elegans* frailty by executing cells before organismal death, we hypothesized that ferroptosis directly impacts on lifespan and may represent an underlying process that contributes to organismal aging. We found that treatment of *C. elegans* with Lip-1 markedly extended lifespan [*Figure 5A and B*; average ~70% increase in median lifespan (eight independent replicates; $p<0.002$), *Supplementary file 4*]. Dose response is shown in *Figure 5—figure supplement 1A & B*. An alternative ferroptosis inhibitor, ferrostatin (*Dixon et al.,*

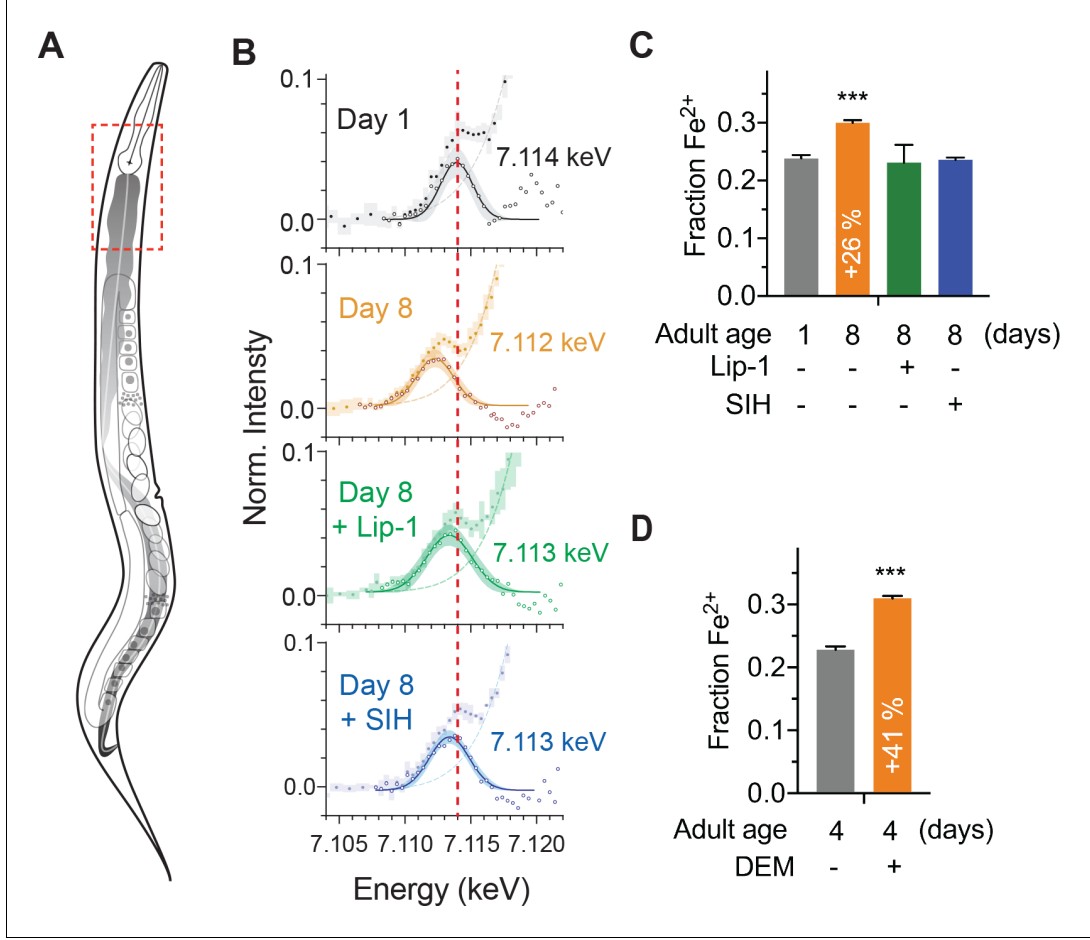

**Figure 4.** Effects of aging, glutathione depletion, SIH and Lip-1 on pro-ferroptotic $Fe^{2+}$ levels. $\varphi$XANES evaluation of the $Fe^{2+}$ fraction in vivo within intact animals. (**A**) Schematic highlighting the anatomy of an adult hermaphrodite *C. elegans* with the intestine shaded in grey. Dashed box is indicative of the region of animals selected for $\varphi$XANES. (**B**) $Fe^{2+}$ synchrotron microscopy. $\varphi$XANES imaging allowed extraction of the normalised Fe K-edge XANES spectra (coloured circles) from the anterior intestine of Day 1 (*n* = 6) and Day 8 control (*n* = 4), SIH treated (*n* = 4) and Lip-1 treated (*n* = 5) worms. Averaged spectra for each group are shown along with 95% confidence intervals (shading). The pre-edge region, following subtraction of the rising edge (dashed line), highlights changes in the intensity and position of the 1 s → 3d transition. The extracted data (empty circles) and fitted Gaussian (solid lines; shading represents the 95% CI) are superimposed to determine the centroid values for the pre-edge peak, from which the $Fe^{2+}$ fraction is derived. Changes in the first derivative of the Fe K-edge XANES (***Figure 4— figure supplement 2***) reflect variation in the intensity of the 1 s → 4 s and 1 s → 4 p transitions. The relative intensity of these features was used to estimate the proportion of $Fe^{2+}$ iron in the specimens. For reference, the red line through all spectra denotes the centroid of the Day 1 adults at 7.114 keV. (**C**) The proportional change in fractional $Fe^{2+}$ contribution for spectrum in the aged (Day 1 versus Day 8 adults) and treated (Lip-1 and SIH, from panel B) specimens is indicated, along with 95% confidence interval. Changes in the first derivative of the Fe K-edge XANES (***Figure 4—figure supplement 2***) was used to infer variation in the intensity of the 1 s → 4 s and 1 s → 4 p transitions and the relative intensity of these features was then used to estimate the proportion of $Fe^{2+}$ iron. (**D**) The proportional change in fractional $Fe^{2+}$ contribution for Day 4 adults treated with (*n* = 4) and without acute glutathione depletion via DEM (*n* = 4) is indicated, along with 95% confidence interval.

The online version of this article includes the following source data and figure supplement(s) for figure 4:

**Source data 1.** Data for XANES comparisons.
**Figure supplement 1.** XFM maps of Fe and regions-of-interests for φ-XANES analysis (dashed boxes).
**Figure supplement 2.** Summary of pooled spectra for young (*n* = 6), aged (*n* = 4) TJ1060 animals and aged animals with SIH (*n* = 4) or Lip-1 treatment (*n* = 5) (*i.e.*the mean for all pixels in all ROIs within each group) from scanning the iron K-edge, including features present in the pre-edge (~7.112 keV) and rising edge (~7.124 keV) are shown in **A**, with their corresponding first-derivatives (**B**).

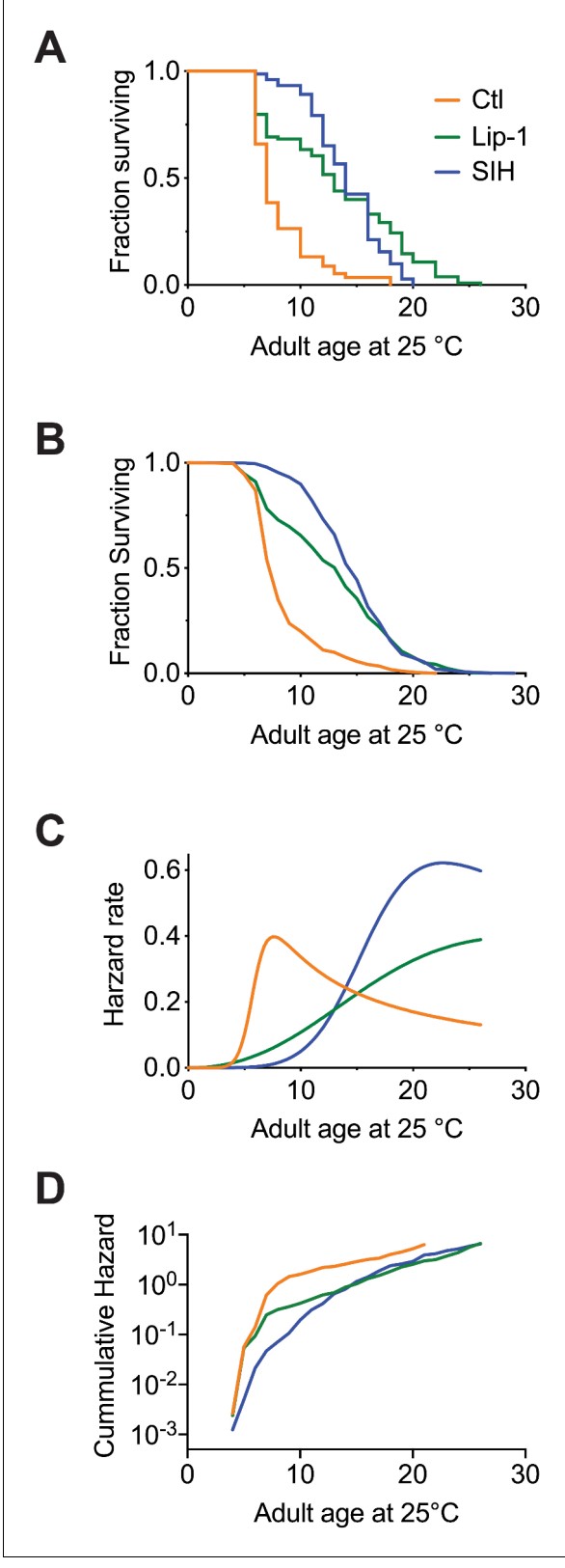

**Figure 5.** Inhibiting ferroptosis extends *Caenorhabditis elegans* lifespan. Treatment with both Lip-1 and SIH extend lifespan. (**A**) Representative Kaplan-Meier survival curve from *C. elegans* treated with vehicle control (Ctl, median survival 7 days, death events $n = 88$); Lip-1 (median survival 13 days, Log-rank test p<0.001, $n = 103$) and SIH (median survival 14 days, p<0.001, $n = 71$) at 25°C. (**B**) Survival curve derived from pooled data from all eight

*Figure 5 continued on next page*

*Figure 5 continued*

replicate experiments (Ctl *n* = 709, Lip-1 *n* = 809, and SIH *n* = 720; see *Supplementary file 4*) at 25°C. (**C**) Plot of hazard (mortality) rate against age at 25°C, derived from meta-analysis of pooled data (presented in B). Both SIH and Lip-1 alter mortality rates relative to control populations and are also distinct from each other. (see *Supplementary file 5*).

The online version of this article includes the following source data and figure supplement(s) for figure 5:

**Source data 1.** Data for survival comparisons.
**Figure supplement 1.** Dose response of SIH, Liproxstatin-1 and Ferrostatin-1.
**Figure supplement 2.** Antibiotic test.

---

*2012*), was also examined, producing a significant but more modest median lifespan extension (*Figure 5—figure supplement 1C*). Targeting the accumulation of late life iron using SIH also resulted in a marked increase in median lifespan [*Figure 5A and B*; average ~100% median increase (eight independent replicates; p<0.0001), *Supplementary file 4*]. Dose response is shown in *Figure 5—figure supplement 1D*. Exposing *C. elegans* to 250 µM SIH as an iron complex (Fe(SIH)$_2$NO$_3$) neutralized the benefits of SIH on lifespan (*Figure 5—figure supplement 1E*), confirming that the rescue mechanism required SIH being free to ligate iron.

## Lifespan increases are not due to temporal scaling

Lip-1 and SIH had distinct effects on aging, as shown by the lifespan curves in *Figure 5*. Treatment with Lip-1 primarily altered late life survival, while SIH extended mid-life with a squaring of the survival curve. Interventions that increase lifespan in *C. elegans* are not uncommon, but Stroustrup et al. recently demonstrated the great majority of longevity interventions, for example dietary and temperature alteration, oxidative stress, and genetic disruptions of the insulin/IGF-1 pathway (*e.g. daf-2* and *daf-16*), heat shock factor *hsf-1*, or hypoxia-inducible factor *hif-1*, each alter lifespan by *temporal scaling* - an apparent stretching or shrinking of time (*Stroustrup et al., 2016*). For an intervention to extend lifespan by temporal scaling it must alter, to the same extent throughout adult life, all physiological determinants of the risk of death. In effect temporal scaling arises when the risk of death is modulated by an intervention acting solely on the rate constant associated with a single stochastic process. It is important to note that temporal scaling is determined by statistical analysis rather than subjective assessment, and also that reproducibility of results depends upon adequate sample size (*Petrascheck and Miller, 2017*).

Combining the replicate data from eight independent experiments (*Figure 5B*), we assessed whether Lip-1 and SIH treatment effects can be explained by the temporal scaling model of accelerated failure time (AFT). We found that the lifespan increases were not consistent with the temporal scaling model (p<0.01; *Supplementary file 5*), so the interventions may target previously unrecognized aging mechanisms. For SIH treatment, the risk of death (hazard) in early adulthood was greatly reduced compared to control populations but rose precipitously in late life (*Figure 5C*). In contrast, Lip-1 markedly reduced the rate of mortality in the post-reproductive period (late-life) with early life mortality closer to that seen in untreated populations. These findings are consistent with ferroptotic cell death limiting lifespan in late life rather than being a global regulator (*e.g.* insulin/IGF-1 pathway) of aging. This raises the possibility of targeted intervention with minimal or no metabolic cost.

The sample size in our experiment is much smaller than the lifespan machine experiment undertaken by *Stroustrup et al., 2016* yet the data against temporal rescaling were significant. To minimize the likelihood that our findings are due to either intrinsic bias in our experiment or inflation of effect size (the *Winner's curse phenomenon*) we also examined the effect of temperature on lifespan intervention. *Stroustrup et al., 2016* reported that changing temperature results in simple temporal rescaling of lifespans; our data corroborated this result and showed that SIH still extended lifespan by a similar dimension at both 20 °C and 25 °C (*Supplementary file 5*).

Our results indicate that while iron accumulation may impact many processes that influence aging rate, ferroptosis inhibition predominantly reduces frailty rather than slows a global rate of aging. Notably, *Stroustrup et al., 2016* identified only two other instances among the many lifespan interventions tested in *C. elegans* that modulated lifespan outside a temporal scaling model, namely altered feeding behaviour (*eat-2* mutants) and mitochondrial dysfunction (*nuo-6* mutants)

(*Stroustrup et al., 2016*). Both these mutants express marked developmental variability and reduced fitness.

## Preventing ferroptosis improves fitness and healthspan

Interventions that increase lifespan in *C. elegans* often do so at the detriment of fitness and healthspan (*Jenkins et al., 2004*; *Walker et al., 2000*). Adult body size can inform on fitness; reduced size may reflect a trade-off between longevity and fitness, as typically seen under dietary restriction where the cost of increased longevity can be lowered size, fertility and movement (*Piper et al., 2011*). Distinctly, SIH-treated animals grew substantially larger. Following one day of treatment all animals were of similar body length (*Figure 6A & B*). After 4 days and 8 days of intervention, adult SIH-treated animals were significantly longer compared to similarly aged controls (*e.g.* control 1440 ± 123 µm *versus* SIH 1696 ± 64 µm, means ± SD on Day 8, p<0.001). In addition, SIH induced an increase in body volume between Days 1 and 4, but not thereafter (*Figure 6C*). SIH-treated worms grew to greater volume than both control and Lip-1-treated worms at Day 4, indicating that preventing iron accumulation can improve animal robustness (for all comparisons see *Supplementary file 6*). Lip-1 had no effect on length or volume.

We also examined whether the interventions altered early and total reproductive output when worms were treated from early adulthood/late L4 (as used in the lifespan experiments). Early fertility (first 24 hr) was not altered by either SIH or Lip-1 treatment (*Figure 6D*; p>0.4). Lip-1 treatment resulted in a small decrease in lifetime reproductive output (*Figure 6D*; p<0.05), but SIH had no effect. Early fertility in *C. elegans* is paramount with respect to Darwinian fitness (*Jenkins et al., 2004*; *Walker et al., 2000*), so the reduction in lifetime fertility with Lip-1 treatment is consistent with a mild deleterious effect in early adulthood.

The effects of both interventions on movement parameters were assessed, since peak motile velocity has been previously demonstrated to correlate strongly with *C. elegans* healthspan and longevity and may be considered the best estimate of healthspan (*Hahm et al., 2015*). As expected, control animals showed a steady decline in maximum velocity as they aged (*Figure 6E*). Treatment with SIH or Lip-1 markedly improved the maximum velocity of aging animals (*Figure 6E*), with increases also in distance travelled and mean velocity (*Supplementary file 6* and *Figure 6—figure supplement 1*).

## Discussion

Our findings indicate that combined late-life dysregulation of glutathione and iron may trigger ferroptosis in *C. elegans* and that activation of this cell death signal limits lifespan. We previously determined that as *C. elegans* age, not only does intracellular iron accumulate but the capacity to safely sequester iron in the iron-storage protein ferritin fails. This contributes to the increased cellular fraction of $Fe^{2+}$ that we observed here, leading to increased oxidative load (*James et al., 2016*; *James et al., 2015*), and providing the specific substrate for ferroptosis (*Gaschler et al., 2018*). The observed depletion of glutathione during aging further lowers the threshold for ferroptosis (*Dixon et al., 2012*).

Thus, in normal aging, a decrease in GSH couples with an age-related increase in labile $Fe^{2+}$ to multiply the likelihood of ferroptosis, leading to death when combined changes in iron and GSH reach a critical threshold (*Figure 7*). We find that SIH and Lip-1 increase lifespan by prohibiting ferroptotic death rather than inhibiting aging rate. We find also that late-life ferroptosis is a prominent contributor to age-related frailty. The healthspan benefits of inhibiting ferroptosis confirm that healthspan improvement need not always require a change in global aging but can result from preventing a cause of frailty, raising an exciting conceptual prospect for therapeutic intervention. Further work is needed to determine the potential windows of action for effective intervention.

Iron is critical to a growing organism, yet unnecessary retention might predispose towards increased frailty later in life (*Gems and Partridge, 2013*; *Hare et al., 2015*). A previous study found that iron supplementation of *C. elegans* shortened lifespan but did not measure in vivo iron levels or the fraction of $Fe^{2+}$ (*Valentini et al., 2012*). Our measures of chronic versus acute changes in the fraction of $Fe^{2+}$ help define the organismal limits of buffering capacity at 0.3 (or 30%). Iron deficiency causes deficits in major developmental pathways, but our interventions were delivered in adulthood

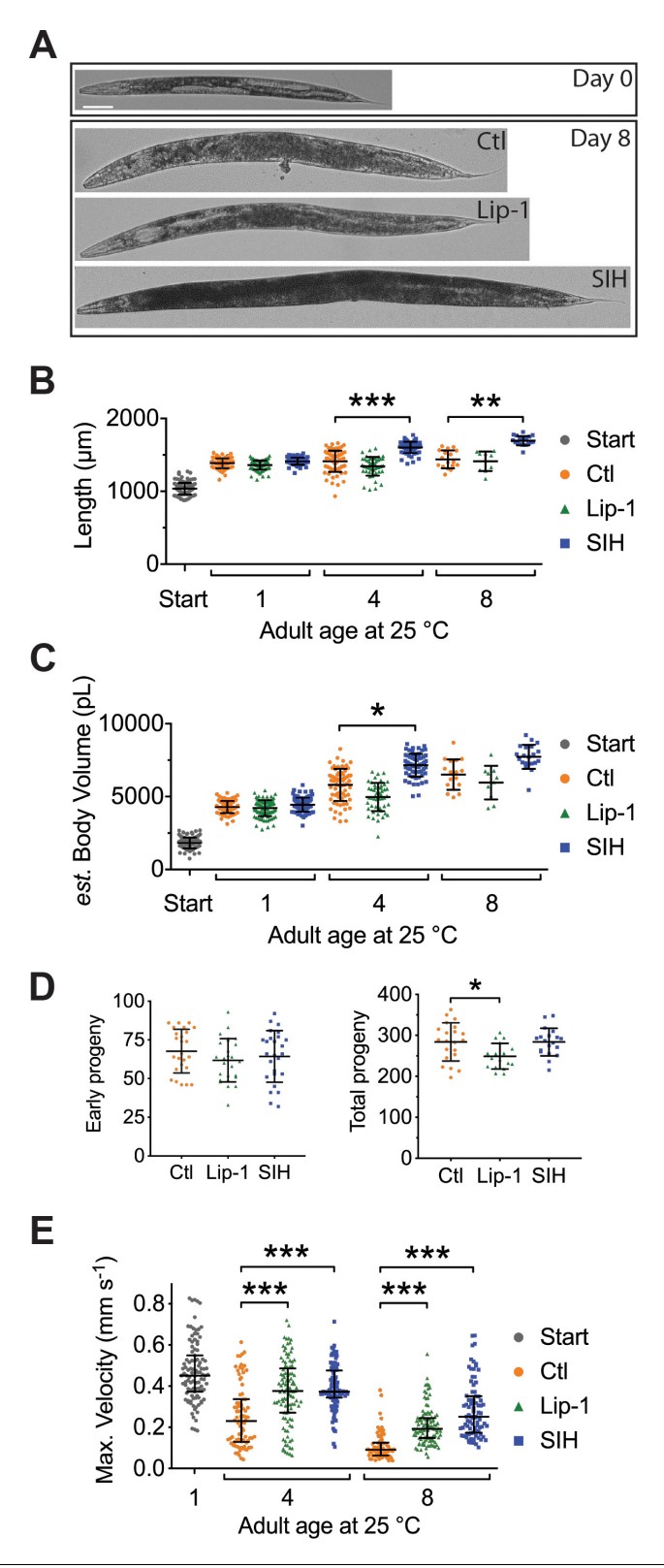

**Figure 6.** Lip-1 and SIH block frailty in *C. elegans*. In all panels, vehicle control (0.5% v/v DMSO, Ctl)-treated worms are shown in orange, Lip-1-treated (200 µM Lip-1) are green and SIH-treated (250 µM SIH) are blue. Significant differences between treatments are highlighted where * indicates p<0.05, ** indicates p<0.01 and *** indicates p<0.001. (**A**) Micrograph of an adult *C. elegans* on the first day of treatment (Day 0) and animals treated

*Figure 6 continued on next page*

*Figure 6 continued*

with Ctl, Lip-1 or SIH for eight days at 25°C (Day 8). Scale bar = 50 µm. (**B**) Estimates of adult body length (in µm), showing SIH- treated animals (blue) have longer average body length compared to age matched control (orange) or Lip-1 (green) treated populations (Kruskal-Wallis ANOVA: H(10) = 432.6, p<0.0001; see *Supplementary file 6* for sample summary and pair-wise comparisons). Start (grey) represents the beginning population of L4/young adults grown from egg at 25°C for 48 hr prior to transfer to treatment plates. Each point represents an individual worm, with mean and error bars representing standard deviation (SD) (**C**) Estimated adult body volume (in pL), showing increased body volume with adult age for all groups (Kruskal-Wallis ANOVA: H(10) = 489, p<0.0001; see *Supplementary file 6* for sample summary and pair-wise comparisons), with SIH treated animals having even greater body volume. Each point represents an individual worm, with mean ± SD. (**D**) Early fertility (first 24 hr) and total reproductive output are unaltered when vehicle control (Ctl)-treated cohorts are compared to Lip-1 or SIH-treated animals at 25°C. Each data point represents an estimate from a single *C. elegans* adult, with mean ± SD. Sample size: early fertility Ctl n = 25, Lip-1 n = 26, SIH n = 26; total Ctl n = 21, Lip-1 n = 17, SIH n = 19 (ANOVA: Early fertility F(2,74)=0.996, p=0.37; Total fertility F(2,57) = 4.89, p=0.011). (**E**) Estimates of maximum velocity achieved by aged and treated cohorts of *C. elegans*. Treatment with either Lip-1 or SIH attenuates the age-related decline in maximum velocity (Kruskal-Wallis ANOVA: H(7) = 298.5, p<0.0001; see *Supplementary file 6* for sample summary and pair-wise comparisons). Each data point represents an estimate from a single *C. elegans* adult, with median ±interquartile range. Equivalent analyses of mean velocity and total distance travelled (and how these data correlate) are shown in *Figure 6—figure supplement 1* and *Supplementary file 6*).

The online version of this article includes the following source data and figure supplement(s) for figure 6:

**Source data 1.** Data for fertility, size and movement.
**Figure supplement 1.** Movement analysis.

to avoid any potential interference with development (*Hare et al., 2015*). This revealed that limiting iron retention post-development was not only tolerable, but improved health and life history traits.

SIH and Lip-1 alter aging at specific life phases and, unlike most known lifespan interventions, do not slow the rate of aging (*Stroustrup et al., 2016*). They act in related but different ways to extend lifespan, with differing impacts on hazard rate and distinct effects on iron levels, life-history traits and acute glutathione depletion. Importantly, both interventions increase both lifespan and health-span without apparent major fitness trade-offs, in contrast to those previously reported in long-lived mutants that do slow the rate of aging (*Jenkins et al., 2004*; *Walker et al., 2000*). Neither SIH nor Lip-1 cause nuclear localization of DAF-16, however, we have previously shown that *daf-2* mutants exhibit markedly attenuated age-related iron accumulation (*James et al., 2015*), so this well-studied lifespan pathway may also impact the rate of ferroptosis. Interestingly, reduction of glutathione via RNAi treatment is reported to decrease *daf-2* lifespan (*Spanier et al., 2010*), but in wild type animals has been found to cause a minor (4–7%) lifespan increase (*Urban et al., 2017*). With respect to lifespan intervention, the non-linear responses observed with glutathione reduction (*Urban et al., 2017*) and compensatory mechanisms (*Ferguson and Bridge, 2019*; *Urban et al., 2017*) suggest that approaches targeting iron and/or lipid peroxidation are likely to be more successful in preventing ferroptosis. The results reported here are consistent with previous orthogonal studies where long-lived genetic mutants of the insulin-like signaling pathway in *C. elegans* express a delay in somatic iron accumulation (*James et al., 2015*). Conversely, ferritin null *C. elegans* mutants, which lack the redox-silent ferritin $Fe^{3+}$ storage pool, are short-lived and contain higher levels of pro-ferroptotic $Fe^{2+}$ (*James et al., 2016*). Although we have demonstrated target engagement of both compounds, additional experiments could also explore the possibility of off-target effects of both SIH and Lip-1.

A caveat for our interpretations is the possibility that the benefits of SIH or Lip-1 were mediated by changes induced in the bacterial food source. Compound by bacteria interactions can play a significant role in *C. elegans* lifespan interventions (*Scott et al., 2017*). This may be particularly relevant when altering iron levels, with conserved microbiome effects on iron uptake and homeostasis (*Qi and Han, 2018*). Further work is needed to determine the influence of host-microbe interactions on iron dyshomeostasis and ferroptosis in this model.

Aging is the principal risk factor for many major human diseases including cancer and dementia. An ancient biochemical dependence on iron may have established an inevitable liability in late life. Needless iron elevation in somatic tissue has been described in many organisms from *Drosophila* to rodents to humans, particularly in brain (*Massie et al., 1983*; *Massie et al., 1985*; *Ward et al.,*

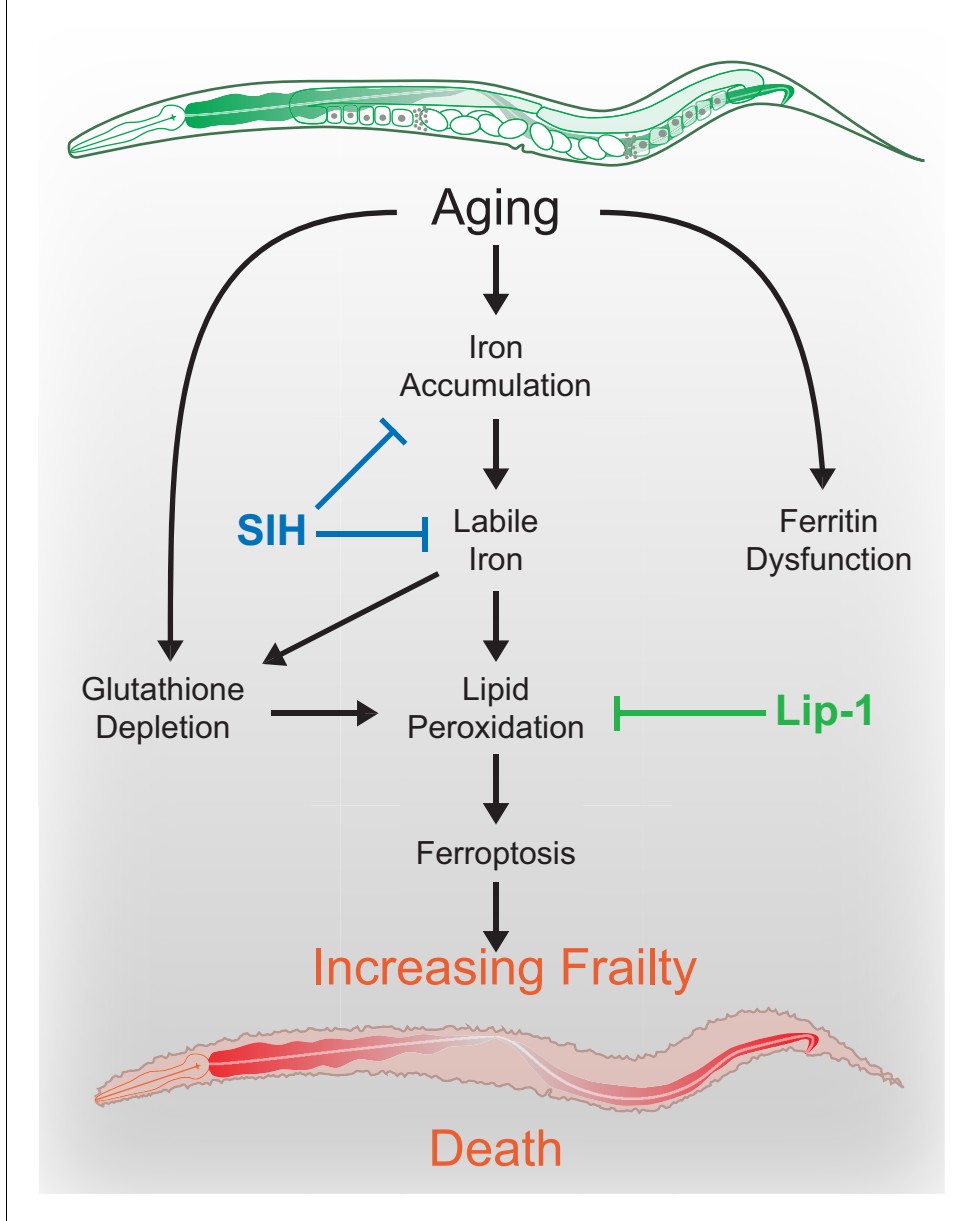

**Figure 7.** Schematic overview. During normal aging iron unnecessarily accumulates. The safe storage of surplus iron in ferritin begins to fail in late life, causing a corresponding elevation of reactive, 'labile' iron. In combination with falling glutathione levels there is increased risk of ferroptotic cell death, via lipid peroxidation signals. These cell death events increase frailty and ultimately shorten organism lifespan. These pharmacological interventions potentially represent targets to improve late life vigour and fitness.

*2014*) and might be a universal feature of aging. Notably, there is no excretion mechanism for systemic iron in animals (*Coffey and Ganz, 2017*; *Muckenthaler et al., 2017*), highlighting that while iron is limiting for development, there has been no evolutionary pressure to regulate its accumulation in post-reproductive life. Ferroptosis plays an important role in kerbing cancer but becomes inappropriately activated in ischemia and neurodegeneration, where its inhibition holds therapeutic promise (*Stockwell et al., 2017*; *Tuo et al., 2017*; *Viswanathan et al., 2017*; *Yang and Stockwell, 2016*). Future studies could test the hypothesis that the cancer-protective benefit of ferroptosis involves the reciprocal acceleration of aging.

# Materials and methods

## Key resources table

| Reagent type (species) or resource | Designation | Source or reference | Identifiers | Additional information |
|---|---|---|---|---|
| Strain, strain background (*Caenorhabditis elegans,* hermaphrodite) | Wild type | *Caenorhabditis* Genetics Center (CGC) | N2 | |
| Strain, strain background (*Caenorhabditis elegans,* hermaphrodite) | *spe-9*(*hc88*); *rrf-3*(*b26*) | *Caenorhabditis* Genetics Center (CGC) | TJ1060 | temperature sensitive -sterile strain |
| Strain, strain background (*Caenorhabditis elegans,* hermaphrodite) | *zIs356* [P*daf-16::daf-16a /b::gfp + rol-6*(*su1006*)] | *Caenorhabditis* Genetics Center (CGC) | TJ356 | DAF-16 reporter strain |
| Antibody | 4-HNE protein adduct antibody (goat polyclonal) | Millipore | Cat.#: AB5605 | (1:2000 dilution) |
| Antibody | anti-Tubulin antibody (mouse monoclonal) | Sigma-Aldrich | Cat.#: T6074 | (1:10,000 dilution) |
| Commercial assay or kit | Thiobarbituric acid reactive substances (TBARS) assay kit | Caymen Chemical | Cat.#: 10009055 | |
| Commercial assay or kit | BCA Protein assay kit | Pierce | Cat.#: 23225 | |
| Chemical compound, drug | Diethyl maleate | Sigma-Aldrich | DEM Cat.#: D97703-100G | |
| Chemical compound, drug | Liproxstatin | Marcus Conrad (Helmholtz Zentrum München; initially) and subsequently ApexBio Tech LLC | Lip-1 | N-[(3-chlorophenyl) methyl]-spiro [piperidine-4,2'(1'H)-quinoxalin]—3'-amine) |
| Chemical compound, drug | Salicylaldehyde isonicotinoyl hydrazone | Des Richardson (University of Sydney) | SIH | |
| Software, algorithm | GeoPIXE 7.3, Dynamic Analysis | CSIRO | | *Ryan, 2000*. Quantitative trace element imaging using PIXE and the nuclear microprobe. Int J Imag Syst Tech 11, 219–230. |
| Software, algorithm | ImageJ WormSizer plugin | | | Moore, B.T., Jordan, J.M., and Baugh, L.R. (2013). WormSizer: high-throughput analysis of nematode size and shape. PLoS One 8, e57142. |
| Software, algorithm | Prism v7.0a, GraphPad Software | | | www.graphpad.com |
| Software, algorithm | R package *flexsurv* | | | rdocumentation.org/packages/flexsurv |
| Software, algorithm | Image J, wrMTrck plugin | | | Nussbaum-Krammer, C.I., Neto, M.F., Brielmann, R.M., Pedersen, J.S., and Morimoto, R.I. (2015). J Vis Exp, 52321 |
| Other | propidium iodide | Life Technologies | P3566 | 0.25 mg/mL solution |

## Strains

Wild type (strain N2), the temperature sensitive-sterile strain TJ1060: *spe-9*(*hc88*); *rrf-3*(*b26*) and the DAF-16 reporter strain TJ356: *zIs356* [P*daf-16::daf-16a/b::gfp + rol-6*(*su1006*)] were obtained from the *Caenorhabditis* Genetics Center. The wild type strain was maintained at 20℃ on standard nematode growth media (NGM) (*Brenner, 1974*) and aged at 20℃ or 25℃ as required. TJ1060 was maintained at 16℃ and also aged at 20℃ or 25℃ as required. TJ1060 was predominately used to remove the inconvenience of progeny production and can be regarded as a proxy for wild type.

## Compounds

Compounds used in this study include

**Chemical structure 1.** Diethyl maleate (DEM) obtained from Sigma-Aldrich.

**Chemical structure 2.** Liproxstatin (Lip-1; N-[(3-chlorophenyl) methyl]-spiro[piperidine-4,2'(1'H)-quinoxalin]—3'-amine) obtained from the laboratory of Marcus Conrad (initially) and subsequently ApexBio Tech LLC.

**Chemical structure 3.** Salicylaldehyde isonicotinoyl hydrazone (SIH) obtained from the laboratory of Des Richardson (University of Sydney).

SIH precomplexed with iron as Fe(SIH)2NO3.

## Glutathione depletion

Diethyl maleate (DEM; Sigma-Aldrich) was added to neat DMSO and added to molten NGM at 55°C to a final concentration of 5, 10, 15 or 20 mM DEM and 0.5 % v/v DMSO. Plates were seeded with OP50 and used within 24 hr. As above, data were collected at 25 (±1) °C using the temperature sensitive-sterile strain TJ1060. A synchronous population was obtained by transferring egg-laying adults to fresh plates at 16°C for 2–3 hr. The adults were removed and the plates with eggs then transferred to 25°C to ensure sterility. After 48 hr at 25°C, when worms were at the late L4/young adult stage, 25–35 nematodes were transferred to fresh plates containing either vehicle control, 250 μM SIH, or 200 μM Lip-1 (all with 0.5 % v/v DMSO). Worms were aged at 25°C for a further 4 days and then transferred to DEM plates. Survival, determined by touch-provoked movement, was scored at 24 and 48 hr after exposure to DEM.

Aging studies were also undertaken to determine changes with age of both survival after DEM exposure and basal glutathione levels. Initial populations were obtained as describe above, with worms aged on standard NGA plates. Note that here we refer to the age of adults as determined by the number of days following the last larval molt and therefore reflects the number of days of adulthood, not the time since egg.

## Quantification of total glutathione

Measurement of total glutathione was based on established protocols and is based on a kinetic spectrophotometric assay using the reaction between GSH and 5,5'-dithio-bis (2-nitrobenzoic acid) (DTNB) measured at 412 nm (*Caito and Aschner, 2015*; *Rahman et al., 2006*). All reagents were freshly prepared prior to the assay and for each estimate 50 adults were collected in 200 μL of S-basal (*Brenner, 1974*) in 1.7 ml microfuge tubes. Animals were washed twice in S-basal, pelleted via centrifugation and total volume reduced to 20 μL. A 50 μL aliquot of Extraction Buffer was

added, then the samples were frozen in Liquid $N_2$ and store at $-80°C$ until required. Extraction buffer consisted of 6 mg/mL 5-sulfosalicylic acid dehydrate, 0.1 % v/v Triton X-100 and Complete, EDTA-free Proteinase inhibitor cocktail (Roche) in KPE buffer (0.1 M potassium phosphate buffer and 5 mM EDTA at pH 7.5).

Samples were homogenized with a Bioruptor Next Gen (Diagenode) bath sonicator, set on HIGH and cooled to 4°C, using 10 cycles of 10 s ON and 10 s OFF. Supernatant was collected following a 14K x $g$ spin at 4°C. Assays were performed in 96 well microplates (clear polystyrene, flat-bottomed, Greiner bio-one), in a total volume of 200 µL per well. To each well was added 50 µL of lysate supernatant, 50 µL of milli-Q $H_2O$ and then 100 µL of GA buffer (NADPH 400 µM, glutathione reductase 1 U/mL and 0.3 mM DTNB in KPE buffer diluent). Reactions were incubated for 1–2 min at room temperature and then absorbance measured at 412 nm for 10 min with 1 min interval using a Powerwave plate spectrophotometer (BioTek). The rate of change in absorbance per minute is linearly proportional to the total concentration of GSH. Total GSH in the samples was interpolated from using linear regression from a standard curve of known GSH concentrations (0 to 1 µM) run in tandem. In parallel, the concentration of total protein per sample was also determined by a Bicinchoninic acid (BCA) assay (Pierce) using the manufacturers protocol. Total GSH estimates were then normalized for protein load, thus accounting for any size differences between populations. Within experiment results are presented as relative glutathione levels, where results are normalized to the mean of the starting population.

## Lipid peroxidation

Measurement of malondialdehyde (MDA) was performed using a Thiobarbituric acid reactive substances (TBARS) assay kit (10009055, Caymen Chemical) as per manufacturer instructions using reduced reaction volumes of 1 mL. For *C. elegans* samples with acute glutathione depletion, Day 1 adults were treated with and without 20 mM DEM for 6 hr at 25°C prior to collection. For aging, animals were aged at 25°C and treated with Lip-1 or SIH as previously described. Replicate samples were collected, washed twice in S-basal, pelleted by centrifugation. Following removal of excess buffer samples (~40 µL) were frozen in liquid-$N_2$ and stored at $-80°C$ until needed. Samples were then homogenized via a Bioruptor bath sonicator (Diagenode, set on 'high power' with 10 cycles of 10 s pulses with a 10 s pause between pulses, at 4°C), then centrifuged at 21,500 x$g$ at 4°C for 30 min and the supernatant retained. The concentration of protein was determined using a BCA assay kit (Pierce) and equivalent aliquots of 20–25 µg total protein used for subsequent measurements.

Analysis of Hydroxynonenal (4-HNE) protein adducts was also used as a proxy for lipid peroxidation. Duplicate samples of 50 and 200 worms were collected and washed twice in S-basal, pelleted by centrifugation and the supernatant discarded. These samples (~30 µL) were frozen in liquid-$N_2$ and stored at $-80°C$ until needed. To each sample an 10 µL 4x Bolt LDS sample buffer (Invitrogen) and 3 µL TCEP (Invitrogen) was added and the sample heated to 95°C for 10 min. Lysates were loaded onto NuPAGE 4–12% Bis-Tris acrylamide gels (1.0 mm, 10-well, Invitrogen), electrophoresed with MES running buffer and then transferred onto 0.45 µm PVDF membrane by electroblot using a Mini Blot module (Invitrogen). 4-HNE protein adducts were detected by an anti 4-HNE protein adduct antibody (1:2000, AB5605, Millipore) in Tris-buffer saline with 5% skim milk, and ECL (GE Healthcare). The membranes were stripped using a 1x ReBlot Strong Antibody Stripping Solution (Merck) for 15 min, reprobed for tubulin using an anti-Tubulin antibody (1:10,000, T6074, Sigma-Aldrich).

## Visualization of cell death

The red-fluorescent propidium iodide (PI), was used to visualize dead cells within live *C. elegans* after DEM treatment and during aging. Populations were incubated for 24 hr at 25°C with PI (a 10 µL volume of 0.25 mg/mL solution added to the bacterial lawn on 50 mm NGM plates) prior to the described age or with concurrent exposure to 10 mM DEM (as described above) and PI. For aging experiments, animals were visualized at Day 6 and Day 8. Cohorts of live animals (*i.e.* showing spontaneous or touch-provoked movement) were isolated and mounted under glass coverslips on 2% agarose pads without anesthetic. Images were captured with a Leica DMI3000B inverted microscope, DsRed filter set and a DFC 3000G digital.

## Liquid chromatography-inductively coupled plasma mass spectrometry

Liquid chromatography was performed using established protocols (*James et al., 2015*). Briefly, samples of aged *C. elegans* were lysed using a Bioruptor Next Gen (Diagenode) bath sonicator set on HIGH and cooled to 4°C using 10 cycles of 10 s ON and 10 s OFF, in a 1:1 vol ratio of Tris-buffered saline (pH 8.0) with added proteinase inhibitors (EDTA-free; Roche). Sample homogenization was confirmed by microscopic inspection. Lysates were then centrifuged for 15 min at 175,000 *g* at 4°C. The supernatant was removed and total protein concentration in the soluble fraction was determined using a NanoDrop UV spectrometer (Thermo Fisher Scientific) before being transferred to standard chromatography vials with polypropylene inserts (Agilent Technologies) and kept at 4°C on a Peltier cooler for analysis. Size exclusion chromatography-inductively coupled plasma-mass spectrometry was performed using an Agilent Technologies 1100 Series liquid chromatography system with a BioSEC 5 s column (5 µm particle size, 300 Å pore size, I.D. 4.6 mm, Agilent Technologies) and 7700x Series ICP-MS as previously described (*Hare et al., 2016b*). A buffer of 200 mM $NH_4NO_3$ was used for all separations at a flow rate of 0.4 mL min$^{-1}$. A total of 50 µg of soluble protein was loaded onto the column by manually adjusting the injection volume for each sample. Mass-to-charge ratios (*m/z*) for phosphorus (31) and iron (56) were monitored in time resolved analysis mode.

Plots of the mean (± standard deviation) of three independent biological replicates are shown. Integration of the three major peaks was performed using Prism (ver. seven for Mac OS X, Graphpad).

## X-ray fluorescence microscopy

### Sample preparation - Elemental mapping

Specimens were prepared for XFM using previously described protocols (*Hare et al., 2016a*; *James et al., 2013*). Briefly, adult *C. elegans* were removed from NGM, washed four times in excess S-basal (0.1 M NaCl; 0.05 M $KHPO_4$ at pH 6.0), briefly in ice-cold 18 MΩ resistant de-ionized $H_2O$ (Millipore) and twice in ice-cold $CH_3COONH_4$ (1.5 % w/v). Samples were transferred onto 0.5 µm-thick silicon nitride ($Si_3N_4$) window (Silson), excess buffer wicked away and then the slide was frozen in liquid nitrogen ($N_2$)-chilled liquid propane using a KF-80 plunge freezer (Leica Microsystems). The samples were lyophilised overnight at −40°C and stored under low vacuum until required.

## Elemental mapping

The distribution of metals was mapped at the X-ray Fluorescence Microscopy beamline at the Australian Synchrotron (*Paterson et al., 2011*) using the Maia detector system (*Kirkham et al., 2010*). The distribution of elements with atomic number < 37 were mapped using an incident beam of 15.6 keV X-rays. This incident energy allowed clear separation of X-ray fluorescence (XRF) peaks from the relatively intense elastic and inelastic scatter. The incident beam (~1.71 109 photons s−1) was focussed to approximately 2 × 2 µm2 (H × V, FWHM) in the sample plane and the specimen was continuously scanned through focus (1 mm sec−1). The resulting XRF was binned in 0.8 µm intervals in both the horizontal and vertical giving virtual pixels spanning 0.64 µm2 of the specimen probed with a dwell time of 8 µsec. XRF intensity was normalized to the incident beam flux monitored with a nitrogen filled ionization chamber with a 27 cm path length placed upstream of the focusing optics. Three single-element thin metal foils of known areal density (Mn 18.9 µg cm−2, Fe 50.1 µg cm−2 and Pt 42.2 µg cm−2, Micromatter, Canada) were used to calibrate the relationship between fluorescence flux at the detector and elemental abundance. Dynamic Analysis, as implemented in GeoPIXE 7.3 (CSIRO), was used to deconvolve the full XRF spectra at each pixel in the scan region to produce quantitative elemental maps (*Ryan, 2000*). This procedure includes a correction for an assumed specimen composition and thickness, in this case 30 µm of cellulose. Though unlikely to exactly match the actual sample characteristics, deviations from these assumptions are not significant for the results presented in this study as the effects of beam attenuation and self-absorption on calcium and iron XRF are negligible for a dried specimen of this type and size (*Davies et al., 2015*).

## Elemental quantification and image analysis

Analysis of elemental XRF maps was performed using a combination of tools native to GeoPIXE and ImageJ (*Schneider et al., 2012*). Incident photons inelastically scattered (Compton scatter) from the sample detail the extent and internal structure of individual *C. elegans*. The differential scattering

power of the specimens and substrate allowed individual animals (or parts thereof) to be identified as regions of interest (ROI; *Supplementary file 7*) facilitating analysis of elemental content on a 'per worm' basis. This segmentation of each elemental map was achieved using the histogram of pixel intensities from Compton maps to locate the clusters within the image. ROIs composed of <10,000 pixels were deemed to be so small that their elemental content was not reflective of the elemental content of whole animals and so these were excluded from the analysis. The 'non-worm' region of each scan was used to calculate the value specimen elemental content was distinguishable from background noise, that is the critical value as defined by *Kadachi and Al-Eshaikh, 2012*. The background corrected elemental maps were used to establish the areal densities and the total mass of each element associated with individual ROIs.

## Sample preparation - $\varphi$**XANES Imaging**

Adult *C. elegans* were removed from NGM, washed four times in excess ice-cold S-basal (0.1 M NaCl; 0.05 M KHPO$_4$ at pH 6.0). Samples were transferred onto 0.5 μm-thick silicon nitride (Si$_3$N$_4$) window (Silson), excess buffer wicked away and then the slide was frozen in situ under a laminar stream of 100 °K dry nitrogen (N$_2$) gas.

## $\varphi$**XANES imaging**

The beam energy was selected using a Si(311) double-crystal monochromator with a resolution of ~0.5 eV. $\varphi$XANES imaging was achieved by recording Fe XRF at 106 incident energies spanning the Fe K-edge (7112 eV). Measurement energy interval was commensurate with anticipated structure in the XANES:

> 7000 eV to 7100 eV: 5 × 20.0 eV steps
> 7100 eV to 7105 eV: 5 × 1.0 eV steps
> 7105 eV to 7135 eV: 75 × 0.4 eV steps
> 7135 eV to 7165 eV: 15 × 2.0 eV steps
> 7165 eV to 7405 eV: 1 × 240.0 eV steps
> 7405 eV to 7455 eV: 5 × 5.0 eV steps

As for XFM, $\varphi$XANES measurements used a beam spot ~2 × 2 μm but data were recorded using continuous scanning at 0.2 mm sec$^{-1}$ (binned at 2 μm intervals). Transit time through each virtual pixel was 10 ms and the incident X-ray intensity at 7455 eV was ~1.67 ×10$^{10}$ photons s$^{-1}$. These imaging parameters gave a total dose associated with the $\varphi$XANES measurement estimated at ~5 MGy. This value is commensurate with doses typically delivered during bulk X-ray absorption spectroscopy.

## $\varphi$**XANES analysis**

The XANES spectra from an iron foil (50.1 μg cm$^{-2}$, Micromatter Canada) was measured to monitor the energy calibration of the beamline. The maxima of the first peak in the derivative spectra of the iron foil was subsequently defined as 7112.0 eV. The energy stability of beamline has been determined at <0.25 eV over 24 hr making energy drift over the course of a scan negligible. Consistency of the measured edge positions in conjunction with stability of beam position and flux recorded in ion chambers upstream the specimen position provide confidence that energy stability was high through the duration of the experiment. Small position drifts were aligned by cross-correlation of the calcium map which remains essentially constant throughout the energy series.

XANES probes the density of states on the absorbing atom and reveals electronic and structural details of coordination environment. The aligned $\varphi$XANES image series is stack of images, one per incident energy allowing the XANES of individual cells to be assessed. Previous work has shown that the distribution of calcium is a useful marker for the position of *C. elegans* intestinal cells and we used this information to identify regions of interest in the $\varphi$XANES stack corresponding to anterior intestinal cells. Anterior intestinal cells were chosen due to their consistent and robust iron content (*James et al., 2013*; see Figure 3—figure supplement 2 for the ROIs and iron maps used for the $\varphi$XANES analysis.

As all points on the specimen represent a heterogenous mixture of iron binding species the resulting XANES spectra are admixtures with contributions from all of these components. The technical particulars of the XFM beamline (being primarily designed for elemental mapping) are not

optimised for high resolution spectroscopy and our XANES spectra are relatively sparse. For iron K-edge XANES the abrupt increase in absorption coefficient at the critical threshold obscures the presence of 1 s → 4 s and 1 s → 4 p electronic transitions. *Berry et al., 2003* demonstrated that the relative intensity of these transitions provides the proportional contribution of $Fe^{2+}$ and $Fe^{3+}$ to the XANES and can be assessed by interrogating the first derivative of the XANES spectra.

## Lifespan determination

Lifespan was measured using established protocols (*James et al., 2015*; *McColl et al., 2010*). SIH was dissolved in neat dimethyl sulfoxide (DMSO; Sigma-Aldrich) then added to the molten NGM at 55°C (to a final concentration of 250 µM SIH in 0.5 % v/v DMSO). Lip-1 was dissolved in neat DMSO then added to the molten NGM at 55°C (to a final concentration of 200 µM Lip-1 in 0.5 % v/v DMSO). Media containing equivalent vehicle alone (0.5 % v/v DMSO) was used for comparison. Standard overnight culture of the *Escherichia coli* (*E. coli*) strain OP50 was used as the food source.

Lifespan data were collected at 25 (±1) °C using the temperature sensitive-sterile strain TJ1060 [*spe-9*(*hc88*); *rrf-3*(*b26*)]. A synchronous population was obtained by transferring egg-laying adults to fresh plates at 16°C for 2–3 hr. The adults were removed and the plates with eggs then transferred to 25°C to ensure sterility. After 48 hr at 25°C, when worms were at the late L4/young adult stage, 25–35 nematodes were transferred to fresh plates containing either vehicle control, 250 µM SIH, or 200 µM Lip-1. All plates were coded to allowing blinding of the experimenter to the treatment regime during scoring. Nematodes were scored for survival at 1 to 3-day intervals and transferred to freshly prepared plates as needed (2–5 days).

*C. elegans* are bacteriophores and the *E. coli* (OP50) monoxenic diet can colonize the pharynx and intestine, resulting in death. Consequently, antibiotics are known to extend *C. elegans* lifespan (*Garigan et al., 2002*). In addition, iron chelating compounds, such as EDTA have been reported to have antibiotic properties. We performed a *disk diffusion test* (*Bauer et al., 1966*) on both Lip-1 and SIH and observed no evidence for inhibition of *E. coli* (strain OP50) growth). Furthermore, an additive effect on median lifespan extension was seen when SIH and the antibiotic ampicillin were co-administered to *C. elegans* (*Figure 5—figure supplement 2*), consistent with independent effects on lifespan.

It is well documented that differences are observed between independent measures of lifespan, with micro-environmental factors such as minor temperature fluctuations potentially resulting differences in median and maximum lifespan between replicates (*Lucanic et al., 2017*). After determining the optimal doses of 250 µM SIH and 200 µM Lip-1, respectively, cohorts of nematodes were compared in eight independent replicates. As the number of worms measured is known to influence the likelihood of accurately observing differences in lifespan (*Petrascheck and Miller, 2017*), the starting populations for all treatments within experiments were in excess of 70 individuals. The median and maximum lifespans observed of control and treated populations for these eight replicates are shown in *Supplementary file 4*. As can be seen in this table, the median lifespan of treated populations was always greater than that of control populations, however the magnitude of the difference varied between experiments, with the median lifespan of control populations ranging from 7 to 9 days.

## Body size analysis

A developmentally synchronous population, derived from eggs laid over a 2 hr window, were cultured on NGA media at 25°C for 48 hr, and then as young adult worms were transferred onto three treatment plates for an additional 24 hr. The treatment plates included NGA with 0.5% (v/v) DMSO (vehicle control, Ctl), 250 µM SIH, or 200 µM Lip-1 (as described above).

Cohorts of approximately 100 animals were transferred into a 1.5 ml centrifuge tube containing 400 µL S-basal. Following a brief centrifugation excess S-basal was removed leaving the animals suspended in 50 µL. Animals were euthanised and straightened by a 15 s exposure to 60°C (using a heated water bath). Samples were then mounted between glass slides and a cover slip and immediately imaged. Micrographs were collected using a Leica M80 stereomicroscope and Leica DFC290 HD 3 MP) digital camera. Pixel sizes were defined using a calibrated 25 µm grid slide (Microbrightfield, Inc). Size and shape metrics were extracted from brightfield images were analysed using the WormSizer plugin (*Moore et al., 2013*) for ImageJ.

## Fertility analysis

Wild type (N2) adults (4 day post egg lay) were transferred to fresh plates for 30 min at 20°C to establish a developmentally synchronous population. Adult nematodes were then removed, and eggs were then transferred to 25°C. As with the survival analyses, after 48 hr at 25°C, when worms were at the late L4/young adult stage individual nematodes were transferred to plates containing vehicle control (0.5 % v/v DMSO), 250 µM SIH, or 200 µM Lip-1. After 24 hr, adult worms were transferred to fresh plates and transferred daily until the end of the fertile period. After allowing progeny to develop for 2 days at 20°C, they were then counted to determine daily and total fertility. Early fertility is determined by the number of progeny laid in the first 24 hr period.

## Movement

A developmentally synchronous population, derived from eggs laid over a 2 hr window, were cultured on NGA media at 25°C for 48 hr, and then as young adult worms were transferred onto three treatment plates for an additional 24 hr. The treatment plates included NGM + 0.5% (v/v) DMSO (vehicle control, Ctl), NGM + 250 µM SIH, and NGM + 200 µM Lip-1 (as described above).

Single worms were transferred to a 55 mm NGA assay plate devoid of a bacterial lawn, without a lid, and left to recover from the transfer for 2 min. Movement of the adults was then recorded using a stereomicroscope (Leica M80) with transmitted illumination from below. A 30 s video recording was captured using a 3 MP DFC290 HD digital camera (Leica Microsystems) at a rate of 30 frames per second. Pixel length was calibrated using a 25 µm grid slide (Microbrightfield, Inc). Recorded series were analysed using the wrMTrck plugin (*Nussbaum-Krammer et al., 2015*) for ImageJ (www.phage.dk/plugins) and Fiji (*Schindelin et al., 2012*) (a distribution of ImageJ).

The maximum velocity achieved was expressed as mm per second (as derived from the distance between displaced centroids per second). Additional metrics of movement were determined including mean velocity (mm s$^{-1}$) and (total) distance travelled (mm). These variables were collated in Prism (v7.0a GraphPad Software) and presented as a scatter plot with medians and interquartile range.

## Quantification and statistical analysis

### Areal density and total iron analysis

Areal iron and total body iron data were assessed for normality using a D'Agostino and Pearson test (see *Supplementary file 2*). Based on this analysis a one-way ANOVA was performed followed by a Sidak's multiple comparisons test (as implemented by PRISM; see *Supplementary file 2*).

### Standard lifespan analysis

Kaplan–Meier survival curves were generated and compared via non-parametric log rank tests (Prism v7.0a, GraphPad Software).

### Testing for departure from temporal rescaling

Following the recently published results of *Stroustrup et al., 2016*, and using their supplied code, we determined whether the results observed with both the SIH and Lip-1 interventions were due to temporal scaling of aging. A modified Kolmogorov-Smirnov (K-S) test was applied to the residuals from a replicate-specific accelerated failure time (AFT) model fitted according to the Buckley-James method that uses a nonparametric baseline hazard function. The function *bj* in R package *rms* was used to fit the replicate-specific model with interventions as categorical independent variables. We used the same approach for testing whether the temperature difference results in simple temporal rescaling, with the only difference being using temperature rather than intervention as categorical independent variable in the AFT model. Full details of these analyses are included in *Supplementary file 5*.

### Characterizing departure from temporal rescaling

Parametric survival models with Weibull baseline hazards and Gamma frailty were fitted to replicate-specific data using the R package *flexsurv*. A likelihood ratio test was used to compare models that assume simple temporal rescaling to models that allow varying degrees of departure from temporal rescaling. The best model for each replicate was selected using a likelihood ratio test and the

goodness of fit (GOF) of the best model is evaluated using a chi-square GOF test. To combine data across different replicates, we performed fixed-effect and random-effect meta-analyses for each parameter in the best model (*Supplementary file 5*). Briefly, the fixed-effect meta-analysis estimates were derived using Inverse Variance Weighting (IVW) in which the estimates from each replicate were weighted by the inverse of their variance estimates. The meta-analysis estimates were then calculated simply as the weighted average of estimates from all replicates. The fixed-effect meta-analysis assumes that there is insignificant variation between the estimates of the same parameter across different replicates. The random-effect meta-analysis also derives the estimates by assigning weights to estimates from each replicate, but in this case the weights take into account the variation of estimates across replicates.

The fixed-effects and random-effects meta-analysis estimates are quite similar and in *Supplementary file 5* we can see that the meta-analysis estimates provide the best fit to SIH data and worst for Lip-1 data. Since there is significant between-replicate variation for the majority of the parameters, it is not surprising that the when the meta-analysis estimates are applied to the real data, a chi-square goodness of fit reveals significant lack of fit ($\chi^2_{(3)}$=237.0 for control worms, $\chi^2_{(5)}$=258.0 for Lip-1 and $\chi^2_{(3)}$=49.7 for SIH, all p-values<0.001).

One notable pattern shown in *Supplementary file 5* is that for nearly all replicates, there is more heterogeneity due to unobserved factors among the control worms, as indicated by the negative $\Delta\log(\sigma^2)$ parameter estimates for Lip-1 and SIH data. This heterogeneity is also reflected in a de-acceleration of the hazard function for control worms beyond 7–8 days. This de-acceleration of the hazard function is the main contributor to the crossing behaviour we observe when comparing the survival functions, and it is what causes a violation of the simple temporal rescaling assumption.

## Survival during GSH depletion

For survival with increasing DEM dose response and protection by compounds (Lip-1 and SIH), data were plotted as fraction of animal alive with upper and lower 95% confidence interval, using the Wilson 'score' method (*Wilson, 1927*) using asymptotic variance (*Newcombe, 1998*) and fitted with a sigmoidal curve (Prism). Pairwise comparisons of treated groups versus control at each concentration of DEM was determined using the N-1 chi-squared test (*Campbell, 2007*; *Richardson, 2011*).

## Fertility

Differences in fertility (*i.e.* early and total reproductive output) were assessed using an ordinary one-way analysis of variance (ANOVA), followed by a Tukey's multiple comparison test (as implemented by Prism v7.0a, GraphPad Software).

## Body length and volume analysis

Data of estimated adult body length and volume were initially assessed for normality using a D'Agostino and Pearson test (see *Supplementary file 6*). Based on this analysis, a nonparametric Kruskal-Wallis Analysis of Variance (ANOVA) was performed followed by a Dunn–Šidák test for multiple comparisons (as implemented by Prism v7.0a, GraphPad Software; *Supplementary file 6*).

## Movement analysis

Data of estimated maximum velocity were initially assessed for normality (see *Supplementary file 6*). Based on this analysis a nonparametric Kruskal-Wallis ANOVA was performed followed by a Dunn–Šidák test for multiple comparisons (as implemented by PRISM; *Supplementary file 6*). Mean velocity and total distance travelled were also determined (*Figure 6—figure supplement 1A & B*). Results summaries and comparisons between treatments are shown in *Supplementary file 6*. The data for the three movement parameters were combined across treatments and ages to determine the relationship between the estimated parameters, all were found to be positively correlated (*Figure 6—figure supplement 1C*).

## Cell death analysis

Differences between the proportion of live animals with fluorescently labelled nuclei in control versus Lip-1 and SIH treatment, either aged or exposed to DEM, were compared using a z-test.

### Type I error for statistical hypothesis testing

Unless otherwise stated, all statistical tests are conducted with type I error set at 0.05.

## Acknowledgements

We thank Abdel Belaidi for comments on the manuscript, Nicholas Stroustrup and Walter Fontana for providing their raw data to enable validation of our temporal scaling analysis and acknowledge the Australian Synchrotron. This study was supported by grants from the Australian Research Council to AIB and GM (DP130100357 and DP180101248), University of Melbourne Research Grant Support Scheme and Miller Foundation to GM, and the Victorian Government's Operational Infrastructure Support Program. We thank the Caenorhabditis Genetics Center (CGC) supported by the US National Institutes of Health - Office of Research Infrastructure Programs (P40 OD010440) for providing *C. elegans* strains.

## Additional information

### Competing interests

Ashley I Bush: AIB is a paid consultant for, and has a profit share interest in, Collaborative Medicinal Development Pty Ltd and is an inventor on patent application 15/505,384, 2017, which covers the method of reducing senescence in a mammal by reducing the concentration of non-ferritin iron. Gawain McColl: GM is an inventor on patent application 15/505,384, 2017, which covers the method of reducing senescence in a mammal by reducing the concentration of non-ferritin iron. The other authors declare that no competing interests exist.

### Funding

| Funder | Grant reference number | Author |
|---|---|---|
| Australian Research Council | DP130100357 | Ashley I Bush |
| Australian Research Council | DP180101248 | Gawain McColl |
| University of Melbourne | | Gawain McColl |
| The Miller Foundation Ltd | | Gawain McColl |

The funders had no role in study design, data collection and interpretation, or the decision to submit the work for publication.

### Author contributions

Nicole L Jenkins, Conceptualization, Data curation, Formal analysis, Investigation, Visualization, Methodology, Writing - original draft, Project administration, Writing - review and editing; Simon A James, Data curation, Formal analysis, Investigation, Methodology, Writing - original draft, Writing - review and editing; Agus Salim, Software, Formal analysis, Writing - original draft; Fransisca Sumardy, Data curation, Investigation, Writing - original draft; Terence P Speed, Formal analysis, Investigation, Writing - original draft; Marcus Conrad, Resources, Writing - original draft; Des R Richardson, Resources; Ashley I Bush, Conceptualization, Resources, Funding acquisition, Methodology, Writing - original draft, Writing - review and editing; Gawain McColl, Conceptualization, Resources, Data curation, Formal analysis, Supervision, Funding acquisition, Validation, Investigation, Visualization, Methodology, Writing - original draft, Project administration, Writing - review and editing

### Author ORCIDs

Gawain McColl https://orcid.org/0000-0002-6347-6722

### Decision letter and Author response

Decision letter https://doi.org/10.7554/eLife.56580.sa1
Author response https://doi.org/10.7554/eLife.56580.sa2

# Additional files

## Supplementary files

- Supplementary file 1. Statistical analysis for *Figure 1*.
- Supplementary file 2. Statistical analysis for *Figure 3*.
- Supplementary file 3. Statistical analysis for *Figure 4* and *Figure 4—figure supplement 2*.
- Supplementary file 4. Statistical analysis for *Figure 5*.
- Supplementary file 5. Details for temporal scaling analysis associated with *Figure 5*.
- Supplementary file 6. Statistical analysis for *Figure 6*.

- Supplementary file 7. The figures showing the ROIs associated with each iron map (ROI; white = overlapped/fractional, green = whole/minimally overlapped, red = excluded from analysis). Shown are the masks used to identify and analyse the iron elemental maps of TJ1060 populations at different adult ages ± Lip-1 or SIH at 25°C. (A) Masks for 1 day old adults (starting population) (B) Masks for 4 day old Control adults (C) Masks for 4 day old SIH-treated adults (D) Masks for 4 day old Lip-1 treated adults E: Masks for 8 day old Control adults (F) Masks for 8 day old 250 µM SIH-treated adults (G) Masks for 8 day old 200 µM Lip-1-treated adults.

- Transparent reporting form

## Data availability

All data generated or analysed during this study are included in the manuscript and supporting files.

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
