## [Decision Letter]

**Acceptance summary:**

This manuscripts illuminates an emerging link between iron metabolism, age-dependent iron accumulation, ferroptosis and ageing in *C. elegans*. In an expansion of their previous work, the authors utilize three compounds (diethyl maleate to modulate in vivo glutathione levels, liproxstatin as an antioxidant "ferroptosis inhibitor", and salicylaldehyde isonicotinoyl hydrazine as an iron chelator) as pharmacological modulators in conjunction with advanced synchrotron-based imaging techniques to investigate these links. The experimental results are of high quality and provide new insights into iron-associated redox homeostasis and ferroptosis as modulators of lifespan.

**Decision letter after peer review:**

Thank you for submitting your article "Changes in ferrous iron and glutathione promote ferroptosis and frailty in aging *Caenorhabditis elegans*" for consideration by *eLife*. Your article has been reviewed by two peer reviewers, one of whom is a member of our Board of Reviewing Editors, and the evaluation has been overseen by Jessica Tyler as the Senior Editor. The reviewers have opted to remain anonymous.

The reviewers have discussed the reviews with one another and the Reviewing Editor has drafted this decision to help you prepare a revised submission.

Summary:

This manuscripts describes the use of three compounds to modulate iron-associated redox homeostasis: diethyl maleate to modulate in vivo glutathione levels, liproxstatin as an antioxidant "ferroptosis inhibitor", and salicylaldehyde isonicotinoyl hydrazine as an iron chelator. After treating animals with combinations of these compounds, the authors identify changes in cell death, biomarkers of lipid peroxidation, iron levels, biomarkers of "healthspan", and lifespan. In this, they combine conventional biochemical and behavioral assays with advanced microscopy techniques/synchrotron-based imaging.

The reviewers judged the experimental results as interesting and agreed that the authors identify a clear link between changes in iron homeostasis and systemic changes in the risk of death. However, the reviewers also pointed out several instances where some of the broader and more general conclusions about aging in *C. elegans* are not strictly supported by the data provided or are at least not the only way to interpret these data.

Essential Revisions:

The authors appear to rely on the assumption that their three drugs act only on their expected targets. No orthogonal experiments were performed that could exclude the possibility that off target effects are responsible for experimental outcomes. It would be much more convincing if the authors could additionally make use of mutations or RNAi constructs that alter iron homeostasis and lipid oxidation, and demonstrate that such alternate means of disrupting iron homeostasis produce the same effects on aging as do the drugs. Failing such experiments, some of the claims should be tempered and potential alternative explanations should be more clearly discussed.

*C. elegans* aging is also highly sensitive to perturbations of their bacterial food source (Han et al., "Microbial Genetic Composition Tunes Host Longevity" Cell, 2017), and drug / bacteria interactions are a major confounder in all pharmacological work in *C. elegans*. However, the possibility that the observed effects are mediated partially or fully by such effects are not discussed or excluded. Any experimental evidence in support of a direct effect on *C. elegans*, rather than through effects on OP50 would be useful. Failing such additional data, the possibility of indirect (OP50-mediated) effects should be clearly outlined.The authors further claim that "the animal dies cell by cell, rather in a single event". While the authors find that some intestinal cells are dead in animals on day 6 and 8 of adulthood, this does not exclude the possibility of a larger (rapid) die off later. Furthermore, ablation of some neurons or of the entire germline actually extends lifespan in *C. elegans*, while other cells may be dispensable for normal lifespan. It is true that the authors observe a correlation between decreased cell death and longer life, but this alone is not sufficient evidence that the early cell-death contributes causally to frailty or that decreased cell death results in longer life.

Another key concern is if the mechanism is truly relevant in normal aging or just in relation to GSH depletion. It is important to establish that this mechanism applies to normal aging. The authors should provide further evidence for this (see below) and discuss this more fully.

The lifespan of WT worms appears shorter than would have expected (even at 25 °C). From the data that is presented, it seems that the authors also did lifespan / control experiments at 20 °C. However, from these trials (at 20 °C), only relative scaling data instead of a full lifespan data is presented (Tables 7-9 in Supplementary file 5 and Figure 7 in Supplementary file 5). Importantly, it appears that only one of the interventions (SIH) but not the other (Lip-1) extended lifespan at 20 °C. Again, the authors should show the full data (even if negative?) on Lip-1 at 20 °C, clearly state the facts and discuss them relative to the question if the observed mechanism indeed applies to normal aging.

It would be important to show the actual lifespan data of WT at 20 °C and to show and fully discuss these data (the lifespan observed at 25 °C and at 20 °C, the shape of the hazard functions for the control populations and the relative protective effects under these conditions). How consistent are these data with the notion that the proposed mechanism is relevant during normal aging and that normal aging is modified by treatment?

The authors state that "Ferroptotic cell death limits lifespan in late life". The authors show that their drugs indeed extend lifespan, and they observe that a time-invariant dose schedule produces a time-varying effect on the hazard rate. However, as they have not identified the timing of action of their drug, they cannot be certain at what age ferroptosis influences frailty. An alternative hypothesis is that their drugs act early in life to permanently alter animals in a way that suppress some forms of death (that kill chronologically young animals) while introducing novel causes of death (that kill chronologically old animals). Without excluding such alternate hypotheses, the authors do not seem to be on solid ground with their preferred conclusion.

---

## [Author Response]

Essential Revisions:The authors appear to rely on the assumption that their three drugs act only on their expected targets. No orthogonal experiments were performed that could exclude the possibility that off target effects are responsible for experimental outcomes. It would be much more convincing if the authors could additionally make use of mutations or RNAi constructs that alter iron homeostasis and lipid oxidation, and demonstrate that such alternate means of disrupting iron homeostasis produce the same effects on aging as do the drugs. Failing such experiments, some of the claims should be tempered and potential alternative explanations should be more clearly discussed.

While we cannot exclude off-target effects, the value of our study is on observing the effect of canonical pharmacological reagents that impact the ferroptosis pathway on longevity and other phenotypes. Ferroptosis is rapidly emerging in the regulated cell death literature as a complex and highly-regulated pathway. This paper provides the first description of ferroptosis in *C. elegans*. Ferroptosis is operationally defined by pharmacological manoeuvres such as the ones we use- induction by glutathione depletion and rescue with iron-chelation or lipid peroxide scavengers. While we are working to find ferroptosis checkpoint homologs in *C. elegans* as an important orthogonal corroboration, this will take time. Nonetheless, we have an important orthogonal corroboration for our intervention – the drop of glutathione that occurs with natural aging, rescued by iron chelation.

We did refer to the possibility of off-target effects and noted the following in our manuscript:

‘To further discount possible off-target stress responses elicited by our interventions, we interrogated DAF-16 localization.’

We have now added additional text in the manuscript to further acknowledge the possibility of off-target effects.

‘Although we have demonstrated target engagement of both compounds, additional experiments could also explore the possibility of off-target effects of both SIH and Lip-1.’

With respect to orthogonal experiments and iron homeostasis, the results presented here are consistent with our previously published work on genetic mutants. For example, we have shown that long-lived *daf-2* mutants have reduced age related iron accumulation. In contrast, short lived *daf-16;daf-2* double mutants have increased iron accumulation (James et al., 2015). We have also shown that *ftn-1; ftn-2* double mutant worms that lack the iron storage protein ferritin (and thus altered iron homeostasis), have increased labile iron and reduced lifespan (James et al., 2016).

We have modified the text in the Discussion section of the manuscript to include reference to these supportive orthogonal results.

‘The results reported here are consistent with previous orthogonal studies where long-lived genetic mutants of the insulin-like signaling pathway in *C. elegans* express a delay in somatic iron accumulation (James et al., 2015). Conversely, ferritin null *C. elegans* mutants, which lack the redox-silent ferritin Fe^3+^ storage pool, are short-lived and contain higher levels of pro-ferroptotic Fe^2+^ (James et al., 2016).’

*C. elegans* aging is also highly sensitive to perturbations of their bacterial food source (Han et al., "Microbial Genetic Composition Tunes Host Longevity" Cell, 2017), and drug / bacteria interactions are a major confounder in all pharmacological work in *C. elegans*. However, the possibility that the observed effects are mediated partially or fully by such effects are not discussed or excluded. Any experimental evidence in support of a direct effect on *C. elegans*, rather than through effects on OP50 would be useful. Failing such additional data, the possibility of indirect (OP50-mediated) effects should be clearly outlined.

We appreciate the concern and have elaborated on this in the revision. In the Materials and methods section we already point out:

‘*C. elegans* are bacteriophores and the *E. coli* (OP50) monoxenic diet can colonize the pharynx and intestine, resulting in death. […] Furthermore, an additive effect on median lifespan extension was seen when SIH and the antibiotic ampicillin were co-administered to *C. elegans* (Figure 5—figure supplement 2), consistent with independent effects on lifespan.’

Nevertheless, we appreciate that this is still a caveat and have introduced the following text into the Discussion to reinforce this consideration:

‘A caveat for our interpretations is the possibility that the benefits of SIH or Lip-1 were mediated by changes induced in the bacterial food source. […] Further work is needed to determine the influence of host-microbe interactions on iron dyshomeostasis and ferroptosis in this model.’

The authors further claim that "the animal dies cell by cell, rather in a single event". While the authors find that some intestinal cells are dead in animals on day 6 and 8 of adulthood, this does not exclude the possibility of a larger (rapid) die off later. Furthermore, ablation of some neurons or of the entire germline actually extends lifespan in *C. elegans*, while other cells may be dispensable for normal lifespan. It is true that the authors observe a correlation between decreased cell death and longer life, but this alone is not sufficient evidence that the early cell-death contributes causally to frailty or that decreased cell death results in longer life.

We have edited the manuscript to remove the phrase ‘the animal dies cell by cell, rather in a single event’.

‘Death of individual cells prior to organismal death is consistent with progressive degeneration contributing to the frailty phenotype.’

Another key concern is if the mechanism is truly relevant in normal aging or just in relation to GSH depletion. It is important to establish that this mechanism applies to normal aging. The authors should provide further evidence for this (see below) and discuss this more fully.

We argue that GSH depletion, which is a feature of normal aging (as shown in Figure 1C), sets up conditions where ferroptosis would also be a feature of normal aging. We also observe cell death prior to organismal death during normal aging (Figure 2C and Figure 2—figure supplement 2) as well as with DEM treatment to reduce GSH (Figure 2B and Figure 2—figure supplement 1).

The text included in the Discussion includes the term ‘normal aging’ to provide additional clarity:

‘Thus, in normal aging, a decrease in GSH couples with an age-related increase in labile Fe^2+^ to multiply the likelihood of ferroptosis, leading to death when combined changes in iron and GSH reach a critical threshold.’

The lifespan of WT worms appears shorter than would have expected (even at 25 C).

Our results are consistent with early work using this strain (TJ1060) without the use of antibiotics e.g. Klass, Aging in the Nematode *Caenorhabditis elegans*: Major Biological and Environmental Factors Influencing Life Span Mech Ageing Dev. 1977 Nov-Dec;6(6):413-29. doi: 10.1016/0047-6374(77)90043-4. This paper reports a mean lifespan of 8.9 ± 1.1 days at 25.5 ^o^C.

The lifespan experiments summarised in Supplementary file 4 indicate variability between experiments (consistent with published results) with a median lifespan of control populations between 7 and 9 days.

As noted in the Materials and methods, all of our lifespan experiments report median adult lifespan, rather than from egg lay (from egg lay would add 3 days to all median values). Median values tend to be somewhat lower than mean values as they are not skewed by a long tail in the lifespan distribution.

It is also important to note that this study uses the temperature- sensitive sterile strain, TJ1060, as a proxy for wild type. Lifespan assays were conducted at 25 ± 1 ^o^C *without* the use of antibiotics (the addition of which extends lifespan, as seen in Figure 5—figure supplement 2 – where addition of ampicillin increases lifespan from a median of 9 days to a median of 15 days).

From the data that is presented, it seems that the authors also did lifespan / control experiments at 20 °C. However, from these trials (at 20 °C), only relative scaling data instead of a full lifespan data is presented (Tables 7-9 in Supplementary file 5 and Figure 7 in Supplementary file 5). Importantly, it appears that only one of the interventions (SIH) but not the other (Lip-1) extended lifespan at 20 °C. Again, the authors should show the full data (even if negative?) on Lip-1 at 20 °C, clearly state the facts and discuss them relative to the question if the observed mechanism indeed applies to normal aging.It would be important to show the actual lifespan data of WT at 20 °C and to show and fully discuss these data (the lifespan observed at 25 °C and at 20 °C, the shape of the hazard functions for the control populations and the relative protective effects under these conditions). How consistent are these data with the notion that the proposed mechanism is relevant during normal aging and that normal aging is modified by treatment?

Apologies for the confusion. We do believe that lifespan experiments conducted at 25 ^o^C represent ‘normal aging’. Use of the temperature sensitive sterile strain TJ1060 was done primarily for economic reasons, as experiments conducted at this temperature are both quicker (saving time) and use less resources (cost-effective, as mandated by the Australian Research Council who funded this research).

Lifespan data at 20 ^o^C was not accrued routinely because of the experimental differences with respect to progeny accumulation requiring daily transfer to fresh plates and because of the high cost of Liproxstatin-1.

Lack of data presented with respect to Lip-1 at 20 °C does not indicate that it has been withheld, rather that it has not been performed with sufficient replication for inclusion in an already data-rich submission. We are happy to provide pilot data (shown in Author response image 1) to reassure the reviewers that the compound effects are not limited to either TJ1060, or to 25 °C, but are also observed in N2 wild type at 20 °C.

These results are consistent with the results included in the manuscript, but unfortunately, we do not have sufficient numbers of worms for formal hazard analysis for N2 or Lip-1 at 20 °C.

**Author response image 1. sa2fig1:** Wild type survival at 20 °C.

The authors state that "Ferroptotic cell death limits lifespan in late life". The authors show that their drugs indeed extend lifespan, and they observe that a time-invariant dose schedule produces a time-varying effect on the hazard rate. However, as they have not identified the timing of action of their drug, they cannot be certain at what age ferroptosis influences frailty. An alternative hypothesis is that their drugs act early in life to permanently alter animals in a way that suppress some forms of death (that kill chronologically young animals) while introducing novel causes of death (that kill chronologically old animals). Without excluding such alternate hypotheses, the authors do not seem to be on solid ground with their preferred conclusion.

The text included in quotation marks above did not appear as written in the manuscript. We have a statement in the Results section, in relation to the temporal scaling analysis, that includes a qualifier which indicates that this is working hypothesis rather than a definitive statement:

‘These findings are consistent with ferroptotic cell death limiting lifespan in late life rather than being a global regulator (e.g. insulin/IGF-1 pathway) of aging.’

In addition to the temporal scaling analysis, the synchrotron imaging results indicate that the reduction of iron observed with SIH treatment is occurring by Day 4 (Figure 2), whereas the observed cell death in control population is demonstrated at Day 6 and Day 8 (Figure 1), consistent with our interpretation.

We have included additional text in the Discussion section to indicate alternate hypotheses.

‘Further work is needed to determine the potential windows of action for effective intervention.’